# Should we keep some distance from distancing? Regulatory and post-regulatory effects of emotion downregulation

**Kersten Diers[1], Denise Dörfel**[1]*****, **Anne Gärtner**[1]**, Sabine Schönfeld[1], Henrik Walter[2],
Alexander Strobel[1], Burkhard Brocke[1]**

**1** Faculty of Psychology, Technische Universität Dresden, Dresden, Germany, **2** Division of Mind and Brain Research, Department of Psychiatry and Psychotherapy, CCM, Charité Universitätsmedizin, Berlin, Germany

* denise.doerfel@tu-dresden.de

## Abstract

Emotion regulation is an indispensable part of mental health and adaptive behavior. Research into emotion regulation processes has largely focused on the concurrent effects of volitional emotion regulation. However, there is scarce evidence considering post-regulatory effects with regard to neural mechanisms and emotional experiences. Therefore, we compared concurrent effects of cognitive emotion regulation with effects at different (immediate, short- and long-term) time intervals. In an fMRI study with N = 46 (N = 30 at re-exposure) young healthy adults, we compared neuronal responses to negative and neutral pictures while participants had to distance themselves from or to actively permit emotions in response to these pictures. We investigated the temporal dynamics of activation changes related to regulation in cognitive control brain networks as well as in the amygdala during stimulation (concurrent effects, timepoint 1) and post-stimulation (immediate, timepoint 2), as well as during re-exposure with the same pictures after short (10 minutes, timepoint 3) and long (1 week, timepoint 4) time intervals. At timepoint 1, negative pictures (versus neutral pictures) elicited a strong response in regions of affective processing, including the amygdala. Distancing (as compared to permit) led to a decrease of this response, and to an increase of activation in the right middle frontal and inferior parietal cortex. We observed an interaction effect of time (stimulation vs. post-stimulation) and regulation (distance vs. permit), indicating a partial reversal of regulation effects during the post-stimulation phase (timepoint 2). Similarly, after 10 minutes (timepoint 3) and after 1 week (timepoint 4), activation in the amygdala was higher during pictures that participants were previously instructed to distance from as compared to permit. These results show that the temporal dynamics are highly variable both within experimental trials and across brain regions. This can even take the form of paradoxical aftereffects at immediate and persistent effects at prolonged time scales.

**Data Availability Statement:** Due to the upload limit on OSF, the data and materials have been partitioned. Behavioral data, second level fMRI data, ROI masks, materials such as the

experimental design and paradigms, the preprint and scripts can be found at OSF: https://osf.io/mg5ac/. First level fMRI data is available here: http://dx.doi.org/10.25532/OPARA-120.

**Funding:** KD, SS, AS and BB received funding from the Deutsche Forschungsgemeinschaft (CRC 940, project A5; www.dfg.de). Open Access Funding by the Publication Fund of the TU Dresden. The funders had no role in study design, data collection and analysis, decision to publish, or preparation of the manuscript.

**Competing interests:** The authors have declared that no competing interests exist.

# Introduction

Research on emotion regulation is based on the basic tenet that subjective experience and physiological reactions are susceptible to voluntary regulation efforts, and that this constitutes an adaption to complex and changing environments [1]. While there are different strategies of emotion regulation, according to the Process Model of Emotion Regulation [2], they all have in common that they alter the subjective experience of an emotion by means of (cognitive) top-down influences.

The Process Model of Emotion Regulation [2] distinguishes five strategies of emotion regulation according to the timepoint in the emotion generation process at which they are implemented: Situation selection, situation modification, attentional deployment, cognitive change (often referred to as reappraisal), and response modulation. A recent taxonomy [3] builds on this model and further specifies the terminology (to be) used in past and future emotion regulation research. Each strategy comprises different tactics. For instance, situation selection might be achieved by behavioral avoidance, modification by problem solving, attentional deployment via distraction, and response modulation by suppression of the emotional expression. In this study, we focus on cognitive change, or reappraisal, that has been repeatedly reported as adaptive emotion regulation strategy in terms of both (short-term) effectiveness and relation to psychopathology [3–6]. Reappraisal appears early in the emotion generation process (antecedent-focused) and refers to altering the value of the emotion eliciting stimulus. It has been assumed that reappraisal is advantageous because the emotional response has not fully unfolded and the affective value of an event itself is altered [6,7]. Reappraisal can be further divided into different tactics, as well [3]: Reinterpretation, i.e. changing the meaning of a stimulus [8], and distancing (previously often referred to as detachment) [see also our own work, 9,10]. The meta-analysis by Webb, Miles and Sheeran (4) showed that while reappraisal has been proven effective overall, distancing ($d_+ = 0.45$) was more advantageous than reinterpretation ($d_+ = 0.36$). According to Powers and LaBar [3], distancing can take different forms: Emotion downregulation by taking a perspective a) that is more spatially distant from the stimulus (spatial), b) that is more distant in time (temporal), c) in which the stimulus represents a hypothetical scenario as opposed to reality (hypothetical), or d) by taking an objective perspective (objective). Most (neuroimaging) studies investigating distancing including our own, so far, implemented experimental designs using the objective form: Participants are instructed to take the perspective of a non-involved, objective observer in order to downregulate emotional responses [9,11]. Objective distancing has been shown to effectively downregulate negative emotions and activate brain regions implied in cognitive control [3,4,12], offering an effective form of emotion regulation to investigate neural correlates and temporal dynamics of successful emotion regulation.

At a neural level, emotion regulation, including cognitive change and attentional deployment, is implemented by cortical regions that exercise influence over emotion-generating subcortical regions [13]. Specifically, emotion-generating regions such as the amygdala have been shown to be responsive to both up- and down-regulation; conversely, regions within the dorsal and ventral frontal cortex, the inferior parietal cortex, and the cingulate cortex are regarded as parts of a common cognitive emotion regulation network [12,14,15]. These two systems are assumed to be negatively coupled [16,17], although there have been conflicting findings [18]. Further research has shown that cortical regions are selectively recruited for the regulation of different kinds of stimuli, for example positive vs. negative [19], for different regulation goals such as up- vs. down-regulation [8], and for different of regulation strategies, e.g. distancing, expressive suppression, or distraction [10,17,20,21]. Taken together, this suggests that while a common and potentially indispensable control network of cognitive emotion regulation can

be identified, additional structures are selectively recruited only under certain circumstances, leading to highly context-dependent patterns of co-activation of different brain structures.

The effects of these networks, however, are not consistently stable over time and cannot be fully understood without the consideration of their evolution across multiple time scales. The idea is that regulation processes act differently at distinct temporal stages of the emotion-generating process and even beyond, that is, after the cessation of the stimulus. Most studies so far have focused on concurrent effects of active emotion regulation. However, it has been shown that emotion regulation effects are also observed beyond the active regulation period, that is during the subsequent post-stimulation phase [immediate effects, see for distancing 9]. In detail, the authors showed an immediate increase of amygdala signal once active regulation via distancing had ended. This was described as a rebound, or paradoxical aftereffect. To qualify as such, the signal needs to demonstrate a crossed interaction effect between time and task, that is decreased concurrent amygdala activation during the stimulation (picture presentation) period and increased activation during the immediate post-stimulation period for a comparison of regulation vs no-regulation conditions. This concept has been extended to a number of brain regions and timepoints. Lamke and colleagues [22] observed decreased activation during emotion regulation in the amygdala and in visual cortex regions in the stimulation phase but increased activation in the subsequent post-stimulation phase. Yet, increased activation was observed during both stimulation and post-stimulation period in the dorsolateral prefrontal cortex and other cortical regions, leading to a distinction between reverse and concordant task-rest interactions.

One study [9] demonstrated not only immediate aftereffects, but also effects across an even more extended period of time up to several minutes (short-term effects). Specifically, upon re-encountering previously regulated negative stimuli in a passive viewing task ten minutes later, the authors observed a persisting down-regulation of the amygdala. This re-exposure effect was negatively correlated to the paradoxical aftereffect that was observed immediately after the stimulation phase; i.e., a higher immediate aftereffect was associated with a smaller short-term regulation effect. A similar study [23] investigated concurrent and long-term effects (1 day) of both emotional up- and down-regulation. After one day, lasting experiential and neural effects were only observed for down-regulation of negative feelings via reinterpretation, but not for other tactics (distraction) or other goals of regulation (up-regulation). Further, the lasting neural effects were confined to prefrontal regions, and were not observed in the amygdala. Independently, it has been shown that regulation effects are also present on long-term time scales of more than one day: Amygdala responses remained attenuated after one week for images that had previously been reappraised [using objective distancing, 24]. This, however, was only observed within the amygdala, but not the frontal cortex, and only happened when the images had been presented and reappraised repeatedly. Another example of long-term effects has been established for episodic memory processes: Down-regulation via distancing during encoding led to prefrontal cortical activation during recognition after one year, whereas lack of regulation during encoding led to activation of rather affective brain regions after the same interval [25].

As systematic investigations of the temporal dimension of distancing including several time intervals are still lacking, the present research aims at replicating and extending previous results on the temporal dynamics of distancing, that is its concurrent, immediate, short- and long-term consequences. We implement a full two-by-two design and compare negative and neutral pictures, achieving a joint maximization of both arousal and valence differences under two task conditions: to distance from (regulation) and to actively permit (no-regulation) all emotions that might arise from negative and neutral pictures. We use multiple timepoints to disentangle the temporal dynamics: 1) a stimulation phase to measure concurrent activation

differences between regulation and no-regulation, 2) a post-stimulation phase to investigate immediate aftereffects, 3) a re-exposure phase after 10 minutes (short-term) as well as 4) a re-exposure phase after one week (long-term) to investigate prolonged aftereffects of distancing.

We expected that this experimental design would lead to activation increases within the amygdala in response to negative pictures, as compared to neutral ones, and that these activation increases would be subject to modulation by the concurrent condition: During the stimulation phase (timepoint 1), we expected relatively less activation in the amygdala when participants were instructed to distance from negative pictures as compared to permitting all upcoming emotions. This should be accompanied by activation increases within several cortical regions, in particular within the frontal and parietal cortex as an indicator of processes associated with cognitive emotion regulation, more specifically objective distancing. Besides the replication of these expected canonical results, we sought to investigate possible paradoxical aftereffects within the amygdala: We expected that the lower amygdala activation in the distance condition as compared to the permit condition in the stimulation phase reverses in the post-stimulation phase (immediate aftereffect, timepoint 2). We further aimed at characterizing the amygdala response during re-exposure with the same pictures after short (10 minutes, timepoint 3) and long (1 week, timepoint 4) time intervals [9,22].

## Materials and methods

This study presents data collected within a larger project on neural correlates and individual differences in emotion regulation and its aftereffects (SFB 940 Project A5). Specifically, the project aimed at elucidating the effectiveness and potential costs of volitional emotion regulation. Different cognitive regulation strategies (acceptance, up- and down-regulation) were compared with respect to their behavioral and neural effectiveness, i.e., emotion regulation success, but operationalized along a prolonged time scale. This allowed to examine potential costs of volitional emotion regulation as indexed by paradoxical immediate and delayed regulatory after-effects. Further aims of the overall project were to investigate associations of emotion regulation success with personality traits and genetic polymorphisms. A detailed overview of the project and its subprojects is given in the Supplementary Materials (S1 Fig). Therefore, some of the data reported in this article have been reused in three follow-up studies within the scope of this project: in accordance with the a priori specified analysis plan (http://gepris.dfg. de/gepris/projekt/223659428 and https://tu-dresden.de/bereichsuebergreifendes/sfb940/ research/a-mechanismen/a5), associations with genetic polymorphisms were investigated [26], as well as the relation between emotion regulation and personality [27]. Additionally, associations of emotion regulation success and dispositional emotion regulation with resting-state cortico-limbic connectivity have been analyzed [18]. Results from the present sample on the research questions of this publication have not been reported in any of these publications. We report how we determined our sample size, all data exclusions (if any), all manipulations, and all measures in our study, as recommended by transparency guidelines [28]. All procedures performed in studies involving human participants were in accordance with the ethical standards of the institutional and/or national research committee and with the 1964 Helsinki declaration and its later amendments or comparable ethical standards. The experimental protocol was approved by the ethics committee of the Technische Universität Dresden (EK10012012). Data and materials are provided at the Open Science Framework (https://osf.io/mg5ac/).

### Participants

Sample size was defined based on feasibility considerations. This resulted in a target sample size of over 48 participants. The sample size that we considered feasible to collect, enabled us

with the possibility to at least detect medium-sized effects at a significance level of 0.05 and a power of 0.80. Forty-eight volunteers (16 male, age range 18 – 38 years, age mean ± $SD$ = 24.6 ± 4.1 years) were recruited from the university community. All participants were right-handed and did not report any current or prior neurological or psychiatric illness or treatment. All participants provided written informed consent and received financial compensation. One participant did not complete the experiment, and another participant withdrew from the study. MRI data were therefore available for 46 (out of 48) participants. Out of these 46 participants, 30 returned after 1 week for their second MRI measurement and were included in the analyses of the follow-up measurements (re-exposure, timepoint 4), while 10 returned at an earlier or later time (after 6 to 30 days) and were hence not included in the present analysis, and 6 did not return at all for the second session.

## Experimental paradigm and procedure

The study consisted of two sessions, one week apart [for detailed descriptions see 18,27]. During the first session (60 min), participants performed a preparatory scan (5 min), four runs of an emotion regulation task (36 min), an anatomical scan (8 min) and a re-exposure task (10 min). During the second session (25–35 min), participants performed a preparatory scan (5 min), a resting state measurement (8 min) and repeated the re-exposure task (10 min). Additionally, participants were asked to fill in questionnaires with regard to individual differences in emotion regulation and their subjective experience during the fMRI measurement as well as provided a blood sample, which are not focused on in the present publication [for a detailed description see 26,27].

**Emotion regulation task (timepoints 1 and 2).** During the emotion regulation task, participants were asked to either permit the emotions arising in response to a set of negative and neutral pictures, or to down-regulate them by means of objective distancing. During the "permit" condition, participants were asked to take a close look at the picture and permit any emotions that might arise as a result. They were encouraged to imagine that they were immediately witnessing the depicted situation, and instructed not to voluntarily intensify their emotions, to re-interpret the situation, or to distract themselves. During the "distance" condition, they were asked to "take the position of a non-involved observer, thinking about the picture in a neutral way". This could be achieved, for example, by reducing the personal involvement with the depicted situation, for instance by assuming a personal or physical distance; again, participants were instructed to refrain from interpreting the situation as not real, attaching a different meaning to the situation, or distracting themselves. The "distance" and "permit" instructions were chosen based on previous work, which demonstrated their efficacy [10,29]. All participants received written instructions including examples, completed a training session outside the MR scanner, which took about 10–15 min and consisted of 16 trials, and were subsequently interviewed about how they implemented the proposed emotion regulation tactic.

Each of the four runs of the emotion regulation task consisted of 16 trials, encompassing four trials for each condition. At the beginning of each trial, a picture was presented for 10 s (stimulation phase, timepoint 1). During the initial 2 s of this period, a semi-transparent overlay was presented across the center of the picture, which contained, as a single word, the instruction for either the "permit" or the "distance" condition. Following the offset of the picture, a fixation cross was presented for a variable period of 16–24 s (post-stimulation phase, timepoint 2). This rather long period was inserted into the trial to provide the participants with a relaxation phase, and to allow the return of the BOLD response to baseline levels. Altogether, the total duration of a single trial was, on average, 30 s. At the end of each run, participants were asked to give a rating of their retrospective subjective arousal. For each

experimental condition, participants rated on a continuous scale, how much aroused they felt during the presentation of the negative and neutral images and the "distance" and "permit" instructions, respectively. The scale for the continuous arousal scale ranged from –200 to 200 for technical reasons (screen coordinates). However, the particular choice of scale units and limits did not affect the participants' ratings, since only verbal anchors were provided at either end of the scale and participants had to move a slider to a position between those anchors that resembled their arousal ("not at all aroused" vs. "very much aroused").

**Re-exposure task (timepoints 3 and 4).** The re-exposure tasks at the end of session 1 (timepoint 3, 10 minutes later) and during session 2 (timepoint 4, 1 week later) consisted of the presentation of exactly those negative and neutral pictures that participants had seen during the emotion regulation task. Since we wanted to investigate emotional reactivity, the presentation duration was shortened to 1000 ms, in accordance with the emotional reactivity paradigm in the study by Walter and colleagues [9]. In contrast to the emotion regulation task, each presentation was followed by a variable inter-trial interval of 2 s to 12 s, and participants should passively view the pictures. Specifically, they were instructed not to voluntarily change their emotional experience as they had done during the main experiment.

## Stimuli

Stimuli were selected from the International Affective Picture System [IAPS, 30] and the Emo-PicS picture set [31]. We used two sets of negative pictures and two sets of neutral pictures (16 pictures per set) matched for content, arousal, and valence (mean valence negative pictures: set 1 = 2.71, set 2 = 2.65; mean arousal negative pictures: set 1 = 5.85, set 2 = 5.69; mean valence neutral pictures: set 1 = 5.17, set 2 = 5.13; mean arousal neutral pictures: set 1 = 2.94, set 2 = 2.96). The negative pictures consisted primarily of depictions of animals, bodies, disaster, disgust, injuries, suffering, or violence, while the neutral pictures depicted various scenes, objects and people. The negative and neutral picture sets were matched with regard to depictions of faces, other parts of the body, single or multiple persons, animals, and inanimate objects. The assignment of the pictures to either the "distance" or "permit" conditions was counterbalanced. In order to rule out any further stimulus- or content-related confounds, two sets of negative and neutral pictures were used for one half of the participants, and the other two negative and neutral sets were used for the other half of the participants. Each set of pictures was arranged in two different sequences assigned to participants in an alternating fashion. For the emotion regulation task (timepoints 1 and 2), the order of stimuli was pseudo-randomized within each sequence with the constraint that all experimental conditions appeared equally within each experimental run and that no more than three presentations of the same experimental condition occurred in succession. For the re-exposure tasks (timepoints 3 and 4), the order of stimulus presentation was randomly shuffled. All pictures were projected to a screen located at the rear end of the scanner and were viewed through a mirror attached to the head coil.

## Data acquisition

Magnetic resonance (MR) imaging was done on a 3 Tesla scanner (Siemens Trio; Siemens Erlangen, Germany), using a 12-channel head coil. Functional (T2*) MR images were acquired using an EPI sequence with 42 axial slices (slice thickness 2 mm) per volume (TR 2410 ms; TE 25 ms; flip angle 80˚; slice gap 1 mm; field of view 192 × 192 mm; matrix size 64 × 64). In addition, anatomical (T1) images were acquired using an MPRAGE sequence consisting of 176 sagittal slices of 1 mm thickness (TR 1900 ms; TE 2.26 ms; flip angle 9˚; FOV 256 mm × 256 mm; matrix size 256 × 256).

## Data analysis

**Behavioral data analyses.** Analysis of the subjective ratings and possible relations between subjective and physiological measures was performed with R 3.0.2 (http://r-project.org) including the ggplot2 package [32], and consisted of a two-way repeated-measures ANOVA with the factors picture (negative, neutral) and regulation (permit, distance) with subsequent post-hoc *t*-tests for dependent samples. Due to technical issues and incomplete measurements, ratings were not available from ten participants.

**fMRI analyses.** Imaging data analysis for all tasks was performed using Matlab 7.4 (Math-Works, Natick, MA) and SPM 8 (http://www.fil.ion.ucl.ac.uk/spm/software/spm8). After discarding the first 4 volumes of each run, preprocessing consisted of motion correction, coregistration of individual functional and anatomical data, spatial normalization of the anatomical images to the MNI template, application of the estimated transformation parameters to the coregistered functional images using a resampling resolution of $2 \times 2 \times 2$ mm$^3$, and spatial smoothing of the functional images (FWHM 8 mm).

First-level statistical analysis of the emotion regulation task was performed using a general linear model with the experimental conditions (detailed below) as regressors and six additional motion regressors of no interest. To account for the notion that "rest" is not necessarily a true rest, especially not in emotionally challenging paradigms [22], we included not only the stimulation but also the post-stimulation phase in our model, starting with the offset of each picture and lasting for the same duration as the stimulation phase. This resulted in a first-level model with eight regressors of interest: we modeled the "permit neutral", "permit negative", "distance neutral", and "distance negative" conditions both for the stimulation (onset of picture presentation, timepoint 1) and post-stimulation (offset of picture presentation, timepoint 2) phase. As the temporal dynamics of amygdala activation may differ from those in cortical regions, we conducted an additional sensitivity analysis for the amygdala, where activation was modeled by a stick function (transient response) in addition to a boxcar function (sustained response). This resulted in two different first-level models for the amygdala, whereas a single first-level model was used for all other brain regions. All regressors of interest were convolved with the canonical hemodynamic response function, and the default high-pass filter for SPM8 (128s) was used. The imaging runs of the emotion regulation task were combined within one fixed-effects model. Parameter estimates for the contrasts of interest were averaged across runs, submitted to a second-level, random-effects analysis and evaluated using one-sample *t*-tests.

The emotion regulation task follows a 2\*2\*2 factorial design with the factors "picture" (negative vs. neutral), "regulation" (permit vs. distance), and "time" (stimulation phase vs. post-stimulation phase). For the stimulation phase, we analyzed the main effects of "picture" and "regulation" as well as the interaction of "picture" and "regulation", which were the contrasts of primary interest. Next, we repeated this analysis for the post-stimulation phase. Finally, we included "time" as an additional factor into the model and computed the three-way interaction contrast for "picture", "task", and "time" in order to characterize the changes in emotional regulation from stimulation to post-stimulation. Additionally, the comparisons were restricted to either neutral or negative pictures.

The first-level model of the re-exposure tasks (timepoints 3 and 4) included four regressors for "negative permit", neutral permit", "negative distance", and "neutral distance" (assignment to these conditions was based on the preceding emotion regulation task). The duration of all events was set to 1000 ms as this was the duration of picture presentation. For statistical analysis, contrasts were computed for the main and interaction effects of the "picture" and "preceding regulation" conditions, and evaluated in a second-level analysis using one-sample t-tests.

Based on our a priori hypotheses, we employed two regions of interest (ROI), the left and right amygdala as defined by the Harvard-Oxford Subcortical Structural Atlas within the FSL software package (https://fsl.fmrib.ox.ac.uk/fsl/fslwiki/Atlases). Masks were created from the probabilistic segmentations at a threshold of 50%. For statistical analyses within the amygdala, we applied a threshold of $p = .05$ FWE after correction for small volume. For all other analyses, a voxel-wise threshold of $p = .05$ FWE across the whole brain was applied. Activations were labeled using the Harvard-Oxford Structural Atlases as well as the Anatomy Toolbox for SPM8 [33].

Since voxel-wise analyses are limited in integrating data across a predefined anatomical structure, we additionally obtained summary measures of activation within the left and right amygdala for all tasks. For this purpose, we extracted parameter estimates for all experimental conditions from the individual first-level analyses using SPM8's spm_summarise() function and further analyzed these data using repeated-measures ANOVAs with the factors picture, strategy, and/or time. In addition, for the emotion regulation task, we extracted the activation time courses from these regions by using the rfxplot toolbox [34] to provide an illustration of the results of the model-based analyses.

## Results

### Behavioral analysis of the stimulation phase (concurrent effects, timepoint 1)

The retrospective subjective arousal ratings after each run of the emotion regulation task demonstrated an interaction between picture and regulation ($F(1,37) = 16.38$, $p < .001$, $\eta^2 = .016$) as well as main effects of picture ($F(1,37) = 118.12$, $p < .001$, $\eta^2 = .456$) and regulation ($F(1,37) = 107.15$, $p < .001$, $\eta^2 = .160$). On average, negative pictures were associated with higher subjective arousal ratings than neutral pictures. Pictures during "permit" were rated as more arousing than pictures during "distance" and this effect was more pronounced for negative than for neutral pictures. On a scale ranging from –200 to 200, the experimental conditions had the following Mean±SD values: permit negative: 50.1±54.4; distance negative: -17.7±58.1; permit neutral: -75.4±71.0; distance neutral: -112.6±59.2.

### Neuronal activation differences during the stimulation phase (concurrent effects, timepoint 1)

For both, the emotion regulation and the re-exposure tasks, we report the comparison of pictures ("negative > neutral" and vice versa) and regulation tactic ("distance > permit" and vice versa) as well as the interaction effects between picture and regulation. During the emotion regulation task, we consider these effects for both the stimulation phase (timepoint 1) and the post-stimulation phase (timepoint 2), separately. For simplicity, we only report interaction effects between regulation and time (stimulation to post-stimulation phase) in a joint analysis of the stimulation and post-stimulation phase. Regarding the two analytic strategies for amygdala activation (transient and sustained responses), we point out when the choice of the analytical model matters; if not indicated otherwise, results hold for both strategies. Finally, we report effects of picture and regulation during re-exposure after 10 min (timepoint 3) and after 1 week (timepoint 4), separately.

**Activation differences between negative and neutral pictures.** Results of the analysis of activation differences during the stimulation phase are depicted in Fig 1. During the regulation phase, negative pictures elicited greater activation than neutral pictures in a large temporo-occipital cluster as well as a few additional brain regions (Table 1). Specifically, activation peaks were observed in the right fusiform gyrus, the left and right inferior temporal, right inferior frontal, left insula, posterior cingulate cortex as well as in the left and right amygdala.

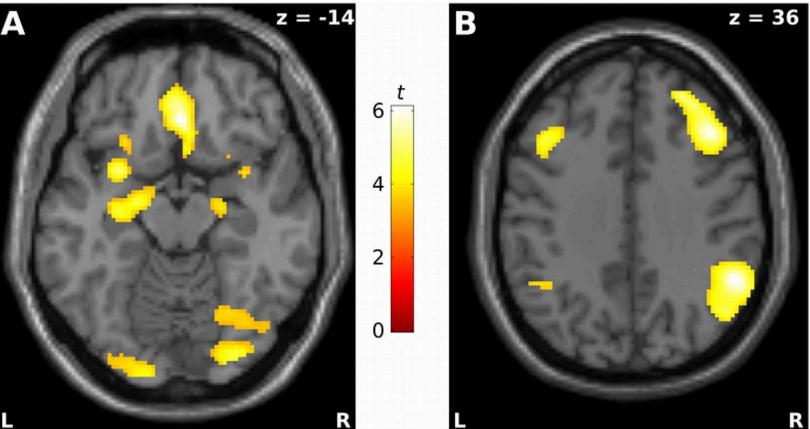

**Fig 1. Whole-brain analysis of concurrent main effects of regulation (stimulation phase, timepoint 1).** A. Permit > Distance during the stimulation phase. B. Distance > Permit during the stimulation phase.

Greater activation during neutral than negative pictures was also observed, primarily in left and right superior temporal as well as left precentral regions.

**Activation differences between distancing and permitting emotions.**   Besides the main effect of picture, there was also a main effect of regulation (Table 1): Clusters of greater activation during distance as compared to permit were observed in the left and right inferior and middle frontal gyrus, left and right inferior parietal lobe and the precuneus, among others. Reverse effects, i.e. greater activation during permit than during distance, were exclusively observed in the left and right amygdala and the occipital cortex. When comparing responses to negative pictures (Table 1), greater activation for distance vs. permit was primarily observed in the right frontal and parietal cortex. Conversely, greater activation for permit vs. distance was present in the left and right amygdala and occipital cortex. Interestingly, similar activation patterns also emerged when only neutral pictures were considered (Table 1).

**Interaction effects between picture and regulation.**   Finally, interaction effects of picture and regulation were observed: the distance vs. permit difference was greater for negative than for neutral pictures in the left and right amygdala for transient responses, while contradictory results were observed for sustained responses (Table 1).

In order to further characterize the effects observed during the voxel-based analysis, we conducted an additional analysis of summary statistics based on the left and right amygdala ROIs (see Fig 2). We observed greater activation for negative than neutral pictures as well as a decrease of this activation during distance vs. permit. That is, there were significant picture and regulation main effects during the stimulation phase, but no interaction effects (left amygdala: picture: $F(1,45) = 30.56$, $p < .001$, $\eta 2 = .112$; strategy: $F(1,45) = 10.08$, $p = .003$, $\eta^2 = .046$; picture × strategy: $F(1,45) = 3.33$, $p = .075$, $\eta^2 = .006$; right amygdala: picture: $F(1,45) = 26.65$, $p < .001$, $\eta^2 = .109$; strategy: $F(1,45) = 9.02$, $p = .004$, $\eta^2 = .037$; picture × strategy $F(1,45) = 1.62$, $p = .210$, $\eta^2 = .003$).

## Neuronal activation differences during the post-stimulation phase (immediate effects, timepoint 2)

Picture and regulation effects were also present during the post-stimulation phase (Table 2): For both, the permit and distance condition, there was a greater activation for negative than neutral pictures in the amygdala ROI, but not in any other brain region. Across picture

**Table 1. Activation maxima in the emotion regulation task during the stimulation phase (concurrent effects, timepoint 1).**

| k | $p_{FWE}$ | t | $p_{unc.}$ | x | y | z | label |
|---|---|---|---|---|---|---|---|
| **Main effect picture** | | | | | | | |
| *Neutral Stim Phase > Negative Stim Phase* | | | | | | | |
| 248 | 0.001 | 6.52 | <0.001 | -32 | -30 | 62 | Left Precentral Gyrus |
| 129 | 0.002 | 6.23 | <0.001 | -58 | -16 | 2 | Left Superior Temporal Gyrus |
| 38 | 0.004 | 6.08 | <0.001 | 58 | -4 | -4 | Right Superior Temporal Gyrus |
| 32 | 0.005 | 5.99 | <0.001 | 12 | -20 | 48 | Right SMA |
| 47 | 0.005 | 5.95 | <0.001 | 44 | -64 | 50 | Right Angular Gyrus |
| 39 | 0.005 | 5.94 | <0.001 | 42 | -14 | 20 | Right Rolandic Operculum |
| 42 | 0.011 | 5.69 | <0.001 | -34 | -20 | 48 | Left Precentral Gyrus |
| *Negative Stim Phase > Neutral Stim Phase* | | | | | | | |
| 6265 | <0.001 | 13.63 | <0.001 | 42 | -46 | -14 | Right Fusiform Gyrus |
| 5965 | <0.001 | 12.28 | <0.001 | -40 | -46 | -16 | Left Inferior Temporal Gyrus |
| 1876 | <0.001 | 10.23 | <0.001 | 44 | 18 | 24 | Right Inferior Frontal Gyrus |
| 1963 | <0.001 | 10.15 | <0.001 | 14 | -20 | -10 | N/A |
| 339 | <0.001 | 7.96 | <0.001 | -32 | 18 | -18 | Left Insula Lobe |
| 134 | 0.001 | 6.62 | <0.001 | -2 | -50 | 26 | Left Posterior Cingulate Cortex |
| 163 | 0.001 | 6.61 | <0.001 | 2 | -56 | -36 | Cerebellar Vermis |
| 25 | 0.006 | 5.91 | <0.001 | 6 | 48 | -20 | Right Gyrus Rectus |
| 36 | 0.016 | 5.56 | <0.001 | 0 | -54 | 44 | Left Precuneus |
| 164 | <0.001 | 6.91 | <0.001 | -24 | -8 | -14 | Left Amygdala (ROI) |
| 183 | <0.001 | 7.70 | <0.001 | 20 | -6 | -12 | Right Amygdala (ROI) |
| 190 | <0.001 | 7.63 | <0.001 | -20 | -8 | -12 | Left Amygdala (ROI, stick model) |
| 211 | <0.001 | 7.86 | <0.001 | 18 | -6 | -12 | Right Amygdala (ROI, stick model) |
| **Main effect regulation** | | | | | | | |
| *Distance Stim Phase > Permit Stim Phase* | | | | | | | |
| 2957 | <0.001 | 9.73 | <0.001 | 36 | 28 | 40 | Right Middle Frontal Gyrus |
| 1967 | <0.001 | 9.65 | <0.001 | 54 | -52 | 38 | Right Inferior Parietal Lobule |
| 274 | <0.001 | 7.35 | <0.001 | -40 | 22 | 34 | Left Middle Frontal Gyrus |
| 302 | <0.001 | 7.07 | <0.001 | 12 | -72 | 38 | Right Cuneus |
| 858 | <0.001 | 6.88 | <0.001 | -42 | -50 | 44 | Left Inferior Parietal Lobule |
| 214 | <0.001 | 6.81 | <0.001 | -12 | -76 | 46 | Left Precuneus |
| 111 | 0.001 | 6.42 | <0.001 | 6 | -38 | 24 | Right Posterior Cingulate Cortex |
| 119 | 0.001 | 6.41 | <0.001 | 66 | -20 | -10 | Right Middle Temporal Gyrus |
| 173 | 0.002 | 6.26 | <0.001 | 36 | 20 | 0 | Right Insula Lobe |
| 129 | 0.003 | 6.23 | <0.001 | -32 | 46 | 8 | Left Middle Frontal Gyrus |
| 71 | 0.003 | 6.17 | <0.001 | 10 | -34 | 42 | Right Middle Cingulate Cortex |
| 70 | 0.009 | 5.80 | <0.001 | 44 | 48 | -4 | Right Inferior Frontal Gyrus |
| 67 | 0.011 | 5.74 | <0.001 | -58 | 24 | 6 | Left Inferior Frontal Gyrus |
| *Permit Stim Phase > Distance Stim Phase* | | | | | | | |
| 159 | 0.001 | 6.43 | <0.001 | -22 | -96 | -8 | Left Inferior Occipital Gyrus |
| 122 | 0.002 | 6.25 | <0.001 | 28 | -94 | 0 | Right Middle Occipital Gyrus |
| 27 | 0.016 | 3.40 | 0.001 | -18 | -10 | -12 | Left Amygdala (ROI) |
| 28 | 0.015 | 3.50 | 0.001 | 18 | -8 | -12 | Right Amygdala (ROI) |
| 122 | 0.001 | 4.33 | <0.001 | -18 | -6 | -12 | Left Amygdala (ROI, stick model) |
| 115 | 0.002 | 4.22 | <0.001 | 22 | -10 | -14 | Right Amygdala (ROI, stick model) |
| **Interaction effect picture X regulation** | | | | | | | |
| *(DistanceNeutral Stim Phase > PermitNeutral Stim Phase) > (DistanceNegative Stim Phase > PermitNegative Stim Phase)* | | | | | | | |

*(Continued)*

**Table 1.** (Continued)

| k | $p_{FWE}$ | t | $p_{unc.}$ | x | y | z | label |
|---|---|---|---|---|---|---|---|
| 13 | 0.044 | 3.05 | 0.002 | -18 | -10 | -12 | Left Amygdala (ROI, stick model) |
| 10 | 0.031 | 3.29 | 0.001 | 32 | -2 | -22 | Right Amygdala (ROI, stick model) |
| *(PermitNeutral Stim Phase > DistanceNeutral Stim Phase) > (PermitNegative Stim Phase > DistanceNegative Stim Phase)* | | | | | | | |
| 5 | 0.023 | 3.33 | 0.001 | 22 | -4 | -26 | Right Amygdala (ROI) |
| **Regulation effect negative pictures only** | | | | | | | |
| *DistanceNegative Stim Phase > PermitNegative Stim Phase* | | | | | | | |
| 1590 | <0.001 | 8.76 | <0.001 | 52 | -48 | 36 | Right Angular Gyrus |
| 2116 | <0.001 | 8.71 | <0.001 | 42 | 26 | 38 | Right Middle Frontal Gyrus |
| 478 | <0.001 | 6.89 | <0.001 | -52 | 38 | -8 | Left Inferior Frontal Gyrus |
| 147 | 0.001 | 6.62 | <0.001 | -42 | 22 | 34 | Left Middle Frontal Gyrus |
| 102 | 0.001 | 6.43 | <0.001 | 62 | -36 | -20 | Right Inferior Temporal Gyrus |
| 294 | 0.003 | 6.21 | <0.001 | -48 | -50 | 44 | Left Inferior Parietal Lobule |
| 113 | 0.004 | 6.11 | <0.001 | 42 | 18 | -10 | Right Insula Lobe |
| 149 | 0.004 | 6.06 | <0.001 | -30 | 44 | 8 | Left Middle Frontal Gyrus |
| 74 | 0.005 | 6.04 | <0.001 | 12 | -66 | 40 | Right Precuneus |
| *PermitNegative Stim Phase > DistanceNegative Stim Phase* | | | | | | | |
| 5 | 0.041 | 3.06 | 0.002 | 18 | -8 | -12 | Right Amygdala (ROI) |
| 148 | 0.001 | 4.68 | <0.001 | -20 | -8 | -14 | Left Amygdala (ROI, stick model) |
| 100 | 0.006 | 3.94 | <0.001 | 22 | -10 | -14 | Right Amygdala (ROI, stick model) |
| **Regulation effect neutral pictures only** | | | | | | | |
| *PermitNeutral Stim Phase > DistanceNeutral Stim Phase* | | | | | | | |
| 205 | 0.001 | 6.45 | <0.001 | 36 | 40 | 32 | Right Middle Frontal Gyrus |
| 192 | 0.002 | 6.36 | <0.001 | 60 | -42 | 30 | Right Supramarginal Gyrus |
| *PermitNeutral Stim Phase > DistanceNeutral Stim Phase* | | | | | | | |
| 267 | 0.001 | 6.69 | <0.001 | -20 | -96 | -6 | Left Inferior Occipital Gyrus |
| 182 | 0.003 | 6.21 | <0.001 | 26 | -88 | 6 | Right Middle Occipital Gyrus |
| 12 | 0.038 | 3.02 | 0.002 | -18 | -8 | -18 | Left Amygdala (ROI) |
| 20 | 0.045 | 3.02 | 0.002 | 20 | -8 | -16 | Right Amygdala (ROI) |
| 12 | 0.014 | 3.50 | 0.001 | -16 | -4 | -14 | Left Amygdala (ROI, stick model) |
| 50 | 0.02 | 3.42 | 0.001 | 18 | -8 | -18 | Right Amygdala (ROI, stick model) |

Abbreviations: k = spatial extent, $p_{FWE}$ = $p$-values corrected for multiple comparisons (FWE), $p_{unc.}$ = uncorrected $p$-values, t = $t$-statistics, x, y, z = MNI coordinates. ROI indicates that an activation peak was observed within the left or right amygdala region of interest.

conditions, greater activation was observed after distance than after permit in brain regions such as the occipital cortex, the precuneus, and also the amygdala (for sustained responses). Similar effects emerged when the comparison was restricted to negative pictures and even neutral picture (primarily for sustained responses). Conversely, we did not observe any activation difference for the reverse contrast, that is there was no greater activation after permit vs. after distance in the post-stimulation phase. No interaction effects between picture and strategy were present during the post-stimulation phase.

## Neuronal interaction effects of regulation (permit, distance) and time (stimulation, post-stimulation)

When time was included as additional factor in the analysis (Table 3), that is, when considering changes from stimulation to post-stimulation phase, interaction effects between regulation

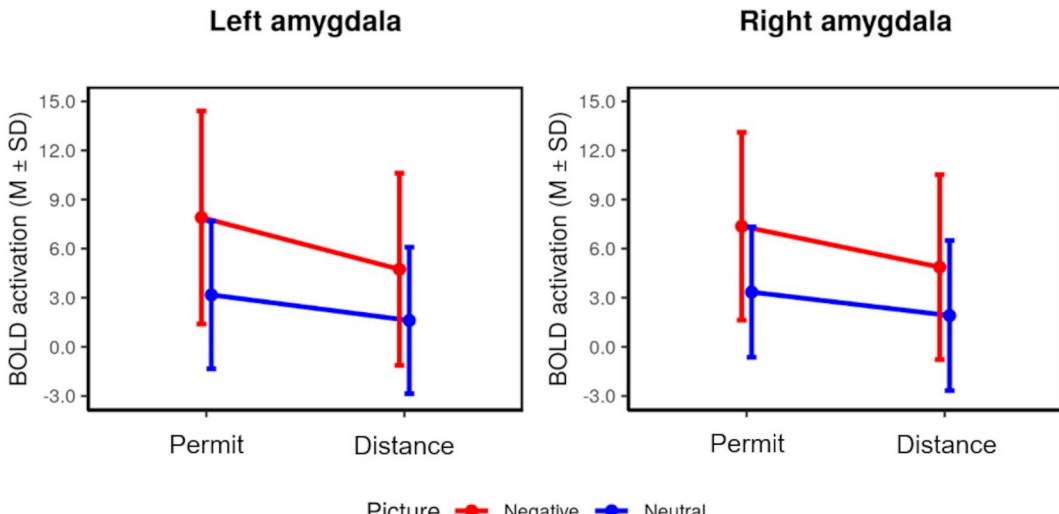

**Fig 2. Summary statistics for the activation of the amygdala ROI during the stimulation phase (concurrent effects, timepoint 1).**

and time were observed in several regions. In the right occipital cortex as well as the left and right amygdala, the effect permit vs. distance was greater during stimulation than during post-stimulation. Conversely, in the right superior frontal gyrus and the right inferior parietal lobe, the reverse effect, that is distance vs. permit, was greater during stimulation than during post-stimulation. Similar patterns emerged when the analysis was restricted to negative pictures and even neutral pictures (for the latter, only in sustained responses).

This analysis was complemented by an additional analysis of summary statistics based on the left and right amygdala ROIs in order to further characterize the voxel-based effects. Specifically, we conducted an analysis for negative pictures that also took the effect of time into account. We found significant regulation-by-time interactions in the left and right amygdala (Fig 3), indicating greater activation in the permit condition than the distance condition during the stimulation phase, but not during the post-stimulation phase (left amygdala: strategy: $F(1,45) = 2.13$, $p = .151$, $\eta^2 = .008$; Time: $F(1,45) = 0.55$, $p = .461$, $\eta^2 = .004$; strategy × Time: $F(1,45) = 11.86$, $p = .001$, $\eta^2 = .029$; right amygdala: strategy: $F(1,45) = 1.33$, $p = .254$, $\eta^2 = .005$; Time: $F(1,45) = 0.94$, $p = .338$, $\eta^2 = .008$; strategy × Time: $F(1,45) = 11.51$, $p = .001$, $\eta^2 = .025$).

**Activation time-courses during the stimulation- and post-stimulation phase (timepoint 1, timepoint 2).** The temporal course of activation in the left and right amygdala was investigated in a post-hoc analysis (Fig 4). It is noteworthy that in this descriptive analysis, an initial peak during the regulation period was followed by a second peak during the post-stimulation period. While the first peak is most pronounced for the permit negative condition, this is no longer the case for the second peak, which shows roughly similar activation levels for all four conditions.

## Neuronal activation differences during the re-exposure tasks

**Re-exposure after 10 minutes (short-term effects, timepoint 3).** During re-exposure after 10 minutes, picture main effects were observed in the posterior cingulate gyrus, the fusiform gyrus, the right temporal gyrus, the left occipital gyrus, and the left amygdala (all negative > neutral, Table 4). There was also a regulation effect (distance > permit) within the left and right amygdala when the analysis was restricted to negative pictures (Fig 5; Table 4). No other effects of interest in this analysis were significant.

**Table 2. Activation maxima in the emotion regulation task during the post-stimulation phase (immediate effects, timepoint 2).**

| k | $p_{FWE}$ | t | $p_{unc.}$ | x | y | z | label |
|---|---|---|---|---|---|---|---|
| **Main effect picture** | | | | | | | |
| *Neutral Post-Stim Phase > Negative Post-Stim Phase* | | | | | | | |
| No results | | | | | | | |
| *Negative Post-Stim Phase > Neutral Post-Stim Phase* | | | | | | | |
| 49 | 0.028 | 3.16 | 0.001 | -22 | -12 | -12 | Left Amygdala (ROI) |
| 27 | 0.033 | 3.16 | 0.001 | 28 | -6 | -14 | Right Amygdala (ROI |
| 75 | 0.004 | 3.94 | <0.001 | -26 | -6 | -18 | Left Amygdala (ROI, stick model) |
| 105 | 0.007 | 3.82 | <0.001 | 24 | -6 | -14 | Right Amygdala (ROI, stick model) |
| **Main effect regulation** | | | | | | | |
| *Distance Post-Stim Phase > Permit Post-Stim Phase* | | | | | | | |
| 75 | 0.004 | 6.03 | <0.001 | -42 | -68 | 4 | Left Middle Occipital Gyrus |
| 46 | 0.006 | 5.87 | <0.001 | -44 | -52 | -24 | Left Inferior Temporal Gyrus |
| 37 | 0.014 | 5.59 | <0.001 | -4 | -76 | 48 | Left Precuneus |
| 53 | 0.024 | 5.39 | <0.001 | -42 | -54 | 54 | Left Inferior Parietal Lobule |
| 18 | 0.028 | 3.12 | 0.002 | -26 | -12 | -14 | Left Amygdala (ROI) |
| 93 | 0.002 | 4.21 | <0.001 | 32 | -2 | -20 | Right Amygdala (ROI) |
| *Permit Post-Stim Phase > Distance Post-Stim Phase* | | | | | | | |
| No results | | | | | | | |
| **Interaction effect picture X regulation** | | | | | | | |
| *(DistanceNeutral Post-Stim Phase > PermitNeutral Post-Stim Phase) > (DistanceNegative Post-Stim Phase > PermitNegative Post-Stim Phase)* | | | | | | | |
| No results | | | | | | | |
| *(PermitNeutral Post-Stim Phase > DistanceNeutral Post-Stim Phase) > (PermitNegative Post-Stim Phase > DistanceNegative Post-Stim Phase)* | | | | | | | |
| No results | | | | | | | |
| **Regulation effect negative pictures only** | | | | | | | |
| *DistanceNegative Post-Stim Phase > PermitNegative Post-Stim Phase* | | | | | | | |
| 6 | 0.026 | 3.15 | 0.001 | -26 | -10 | -16 | Left Amygdala (ROI) |
| 41 | 0.015 | 3.44 | 0.001 | 24 | -8 | -18 | Right Amygdala (ROI) |
| 6 | 0.059 | 2.94 | 0.003 | 24 | -8 | -18 | Right Amygdala (ROI, stick model) |
| *PermitNegative Post-Stim Phase > DistanceNegative Post-Stim Phase* | | | | | | | |
| No results | | | | | | | |
| **Regulation effect neutral pictures only** | | | | | | | |
| *DistanceNeutral Post-Stim Phase > PermitNeutral Post-Stim Phase* | | | | | | | |
| 7 | 0.012 | 3.54 | <0.001 | -32 | 0 | -18 | Left Amygdala (ROI) |
| 49 | 0.033 | 3.16 | 0.001 | 32 | -2 | -22 | Right Amygdala (ROI) |
| *PermitNeutral Post-Stim Phase > DistanceNeutral Post-Stim Phase* | | | | | | | |
| No results | | | | | | | |

Abbreviations: k = spatial extent, $p_{FWE}$ = *p*-values corrected for multiple comparisons (FWE), $p_{unc.}$ = uncorrected *p*-values, *t* = *t*-statistics, x, y, z = MNI coordinates. ROI indicates that an activation peak was observed within the left or right amygdala region of interest.

**Re-exposure after 1 week (long-term effects, timepoint 4).** Re-exposure after 1 week led to greater activation for negative than for neutral pictures in several brain regions, including the right inferior temporal gyrus, left occipital, fusiform, and inferior frontal gyrus, and the left and right amygdala (Table 5). Again, there was an effect of the previous regulation condition: greater activation for previously distanced pictures was present in the right amygdala (Fig 5). Restricting this comparison to negative pictures did not yield any significant effect, but a small

**Table 3. Immediate aftereffects of emotion regulation (interaction effects of regulation (permit, distance) and time (stimulation/timepoint 1, post-stimulation/timepoint 2).**

| k | $p_{FWE}$ | t | $p_{unc.}$ | x | y | z | label |
|---|---|---|---|---|---|---|---|
| **Interaction effect regulation X time** | | | | | | | |
| *(Distance Stim Phase > Permit Stim Phase) > (Distance Post-Stim Phase > Permit Post-Stim Phase)* | | | | | | | |
| 95 | 0.001 | 6.48 | <0.001 | 60 | -50 | 46 | Right Inferior Parietal Lobule |
| 31 | 0.006 | 5.94 | <0.001 | 20 | 8 | 64 | Right Superior Frontal Gyrus |
| *(Permit Stim Phase > Distance Stim Phase) > (Permit Post-Stim Phase > Distance Post-Stim Phase)* | | | | | | | |
| 148 | 0.003 | 6.22 | <0.001 | -18 | -96 | -10 | Left Inferior Occipital Gyrus |
| 127 | 0.005 | 6.00 | <0.001 | 14 | -100 | 0 | Right Calcarine Gyrus |
| 31 | 0.007 | 5.90 | <0.001 | 0 | 2 | -10 | Subcallosal Cortex |
| 39 | 0.008 | 5.84 | <0.001 | 34 | -86 | -14 | Right Inferior Occipital Gyrus |
| 161 | 0.002 | 4.29 | <0.001 | -18 | -12 | -14 | Left Amygdala (ROI) |
| 226 | <0.001 | 4.97 | <0.001 | 32 | -2 | -20 | Right Amygdala (ROI) |
| 134 | <0.001 | 4.97 | <0.001 | -26 | -12 | -14 | Left Amygdala (ROI, stick model) |
| 144 | 0.001 | 4.64 | <0.001 | 24 | -8 | -18 | Right Amygdala (ROI, stick model) |
| **Interaction effect regulation X time, negative pictures only** | | | | | | | |
| *(DistanceNegative Stim Phase > PermitNegative Stim Phase) > (DistanceNegative Post-Stim Phase > PermitNegative Post-Stim Phase)* | | | | | | | |
| 68 | 0.001 | 6.6 | <0.001 | 18 | 12 | 62 | Right Superior Frontal Gyrus |
| *(PermitNegative Stim Phase > DistanceNegative Stim Phase) > (PermitNegative Post-Stim Phase > DistanceNegative Post-Stim Phase)* | | | | | | | |
| 77 | 0.004 | 3.98 | <0.001 | -24 | -12 | -14 | Left Amygdala (ROI) |
| 163 | 0.003 | 4.12 | <0.001 | 24 | -12 | -12 | Right Amygdala (ROI) |
| 114 | <0.001 | 5.40 | <0.001 | -28 | -10 | -16 | Left Amygdala (ROI, stick model) |
| 146 | <0.001 | 5.22 | <0.001 | 24 | -8 | -18 | Right Amygdala (ROI, stick model) |
| **Interaction effect regulation X time, neutral pictures only** | | | | | | | |
| *(DistanceNeutral Stim Phase > PermitNeutral Stim Phase) > (DistanceNeutral Post-Stim Phase > PermitNeutral Post-Stim Phase)* | | | | | | | |
| No results | | | | | | | |
| *(PermitNeutral Stim Phase > DistanceNeutral Stim Phase) > (PermitNeutral Post-Stim Phase > DistanceNeutral Post-Stim Phase)* | | | | | | | |
| 57 | 0.006 | 5.92 | <0.001 | -20 | -96 | -8 | Left Inferior Occipital Gyrus |
| 69 | 0.013 | 3.49 | 0.001 | -16 | -10 | -16 | Left Amygdala (ROI, stick model) |
| 187 | 0.001 | 4.41 | <0.001 | 32 | -2 | -20 | Right Amygdala (ROI, stick model) |

Abbreviations: k = spatial extent, $p_{FWE}$ = *p*-values corrected for multiple comparisons (FWE), $p_{unc.}$ = uncorrected *p*-values, t = *t*-statistics, x, y, z = MNI coordinates. ROI indicates that an activation peak was observed within the left or right amygdala region of interest.

effect of the same kind appeared for neutral pictures (Table 5). No other effect of interest in this analysis exceeded the whole-brain threshold.

## Discussion

This study presents a systematic investigation of emotional processing and regulation across two picture types and regulation conditions and immediate as well as short- and long-term time scales. Its main results can be summarized as follows: During stimulation (at timepoint 1) negative pictures elicited a strong response in affective regions of the brain, most prominently the amygdala, but also the insula and cingulate cortex. Volitional emotion regulation, as implemented by objective distancing, led to a decrease of this response, and to an increase of activation in the right middle frontal and inferior parietal cortex. During the stimulation phase, distancing-related activation in cortical regions appeared as a sustained response, whereas the reverse effect, that is the down-regulation of the amygdala, appeared mainly as a transient response, in particular for negative pictures. Regarding paradoxical immediate aftereffects in

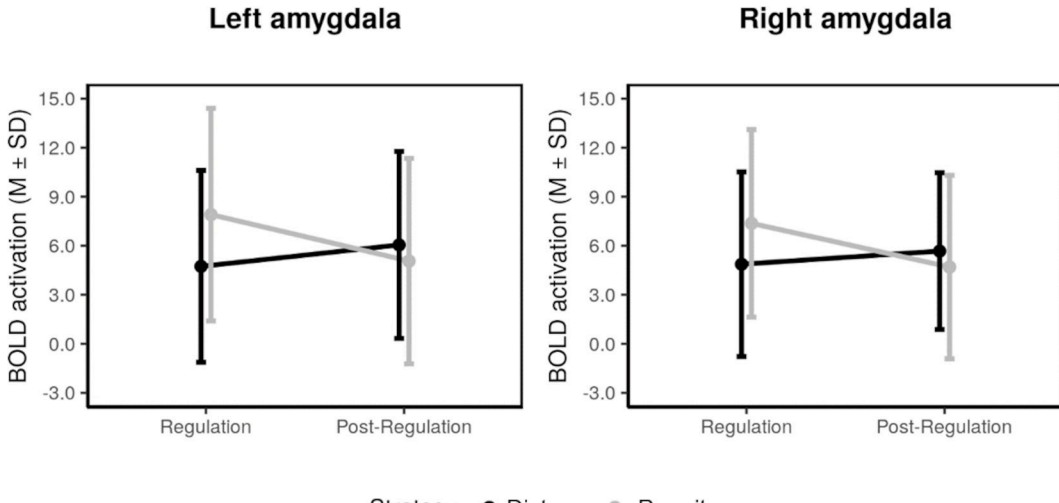

**Fig 3. Summary statistics for the amygdala ROI during the stimulation phase (timepoint 1) and the post-stimulation phase (immediate aftereffects, timepoint 2).** This analysis is limited to negative stimuli.

the amygdala (at timepoint 2), statistically significant interaction effects between time and regulation indicate that higher levels of amygdala activation in the permit condition during the stimulation phase reversed to higher activation in the distance condition during the post-stimulation phase. Further (paradoxical) task-rest interactions for negative pictures were observed in the occipital cortex and the ventromedial frontal/subgenual cingulate cortex. Previous emotion regulation had an impact on amygdala activation both at timepoint 3 (after 10 minutes) and timepoint 4 (after one week): Activation within the amygdala was higher if the participants had previously (in the emotion regulation task) been instructed to distance from the picture as compared to permit all upcoming emotion.

## Effects of emotional processing and regulation at timepoint 1

**Concurrent emotion regulation effects–Replication and extension.** A key result of the present study is the replication of canonical emotion regulation effects, in particular the downregulation of the amygdala and the concurrent activation of a right frontoparietal control

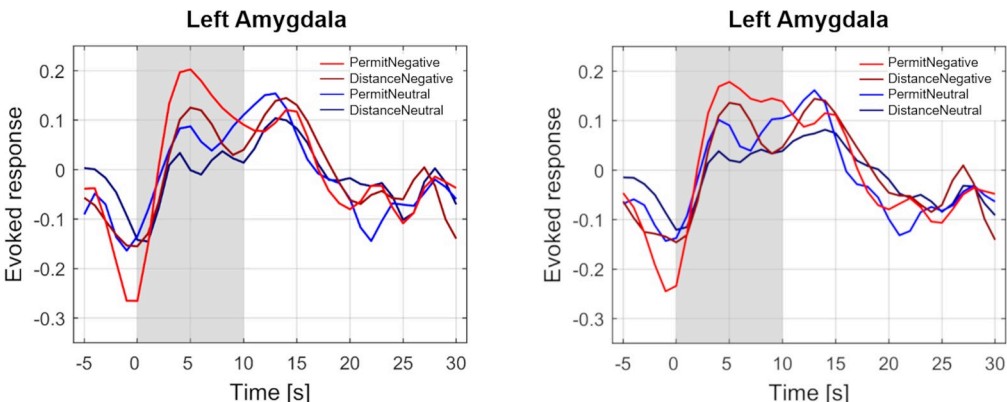

**Fig 4. Activation time-courses in the left and right amygdala during stimulation (timepoint 1) and post-stimulation phase (immediate aftereffects, timepoint 2).** The shaded area indicates the stimulation phase.

**Table 4. Activation maxima during re-exposure after 10 minutes (short-term effects, timepoint 3).**

| k | $p_{FWE}$ | t | $p_{unc.}$ | x | y | z | label |
|---|---|---|---|---|---|---|---|
| **Main effect picture** | | | | | | | |
| *Neutral > Negative* | | | | | | | |
| No results | | | | | | | |
| *Negative > Neutral* | | | | | | | |
| 148 | <0.001 | 8.21 | <0.001 | -8 | -18 | 30 | Left Posterior Cingulate Cortex |
| 428 | <0.001 | 7.96 | <0.001 | 42 | -50 | -16 | Right Fusiform Gyrus |
| 366 | <0.001 | 7.16 | <0.001 | -42 | -48 | -22 | Left Fusiform Gyrus |
| 167 | 0.001 | 6.83 | <0.001 | 50 | -70 | 12 | Right Middle Temporal Gyrus |
| 85 | 0.006 | 6.23 | <0.001 | -44 | -80 | -2 | Left Middle Occipital Gyrus |
| 8 | 0.005 | 3.97 | <0.001 | -16 | -8 | -14 | Left Amygdala (ROI) |
| **Main effect regulation** | | | | | | | |
| *Distance > Permit* | | | | | | | |
| No results | | | | | | | |
| *Permit > Distance* | | | | | | | |
| No results | | | | | | | |
| **Interaction effect picture X regulation** | | | | | | | |
| *(Distance Neutral > Permit Neutral) > (Distance Negative > Permit Negative)* | | | | | | | |
| No results | | | | | | | |
| *(Permit Neutral > Distance Neutral) > (Permit Negative > Distance Negative)* | | | | | | | |
| No results | | | | | | | |
| **Regulation effect negative pictures only** | | | | | | | |
| *DistanceNegative > PermitNegative* | | | | | | | |
| 9 | 0.011 | 3.68 | <0.001 | -20 | -10 | -12 | Left Amygdala (ROI) |
| 43 | <0.001 | 5.74 | <0.001 | 26 | -6 | -14 | Right Amygdala (ROI) |
| *PermitNegative > DistanceNegative* | | | | | | | |
| No results | | | | | | | |
| **Regulation effect neutral pictures only** | | | | | | | |
| *DistanceNeutral > PermitNeutral* | | | | | | | |
| No results | | | | | | | |
| *PermitNeutral > DistanceNeutral* | | | | | | | |
| No results | | | | | | | |

Abbreviations: k = spatial extent, $p_{FWE}$ = $p$-values corrected for multiple comparisons (FWE), $p_{unc.}$ = uncorrected $p$-values, $t$ = $t$-statistics, x, y, z = MNI coordinates. ROI indicates that an activation peak was observed within the left or right amygdala region of interest.

network during distancing from negative pictures. This confirms several previous reports [10,20,35]. This pattern of replications hints at the robustness of our experimental paradigm and can serve as a basis for a systematic extension, as will be discussed below, for the exploration of activation changes in other cortical and subcortical regions, and for the incorporation of additional experimental variables such as time.

A second result of this study is that the observed effects extend beyond the regions that are commonly implicated in emotion regulation. Specifically, negative pictures were not only associated with amygdala and insula activation, but also with activation in the inferior temporal gyrus and the occipital cortex, the latter was also observed during the permit condition. During the distance negative condition, left- (instead of right-) hemispheric activation in the inferior and middle frontal gyrus specifically was an unexpected observation with respect to our hypotheses and previous results [10].

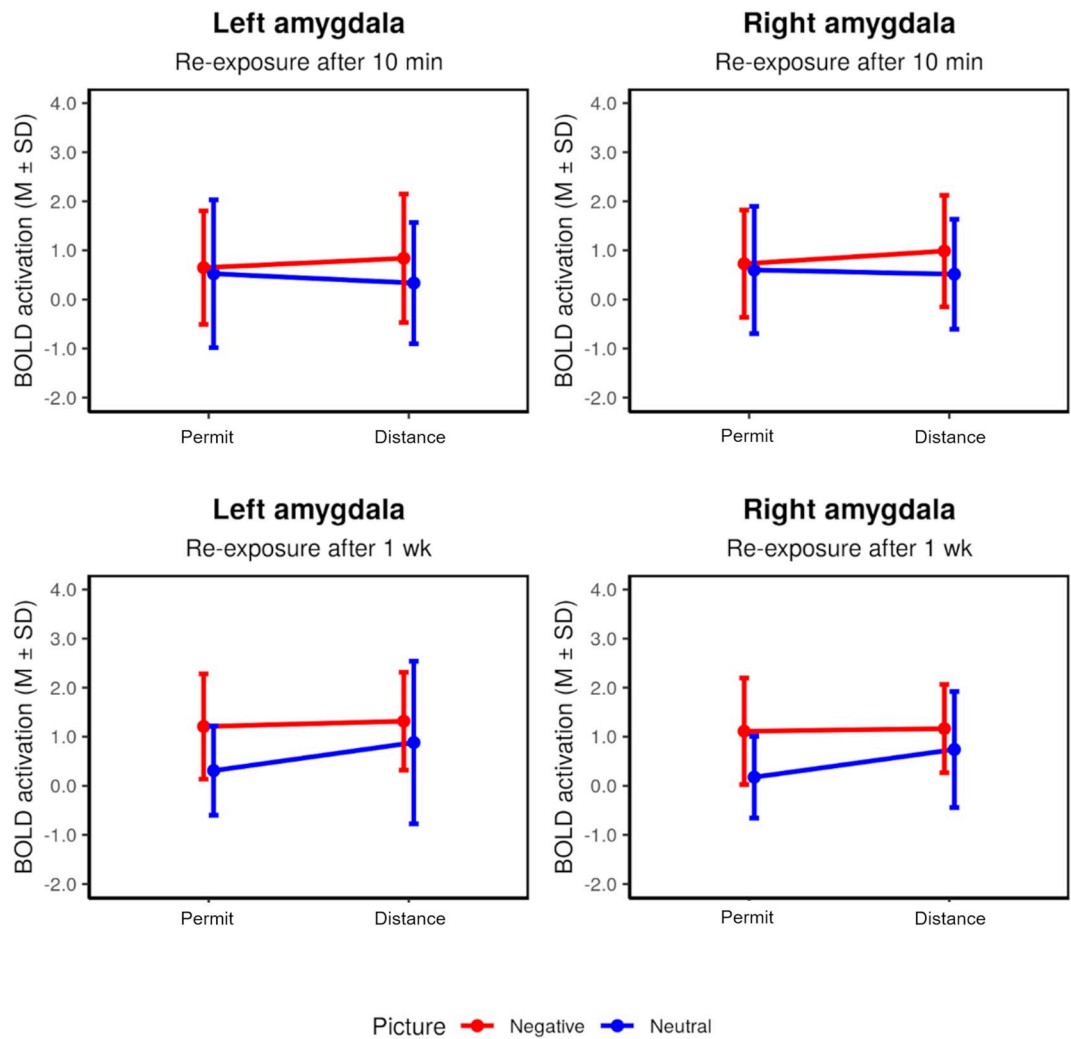

**Fig 5. Summary statistics for the amygdala ROI during re-exposure after 10 min (short-term effects, timepoint 3) and after 1 week (long-term effects, timepoint 4).**

Other authors have noted that emotion regulation–as any complex psychological function–does not rely on single brain regions [for network analyses see for instance 17,36,37]. Emotion regulation is a heterogeneous construct that overlaps to a certain extent with other cognitive domains: for example, Messina, Bianco, Sambin and Viviani [38] distinguish between executive and semantic processes during reappraisal; they suggest that executive functions serve a general role, while semantic regions in the temporal and parietal cortex should be exclusively activated during reappraisal via perspective-taking. With regard to our finding of additional left-hemispheric frontal cortical activation, it is possible that inner speech is a component of distancing via self-instruction that results in selective recruitment of left frontal regions [39] in addition to a more general, right-hemispheric control network. Further support for the idea that higher-level regulation strategies recruit additional neural resources to effectively cope with an emotion-eliciting event is provided by a meta-analysis [40]. This analysis shows that ventromedial prefrontal cortex activation is present in diverse, but related experimental domains such as fear extinction, cognitive emotion regulation (reappraisal), and placebo-mediated changes in negative affect. Only during reappraisal and placebo-mediated changes,

**Table 5. Activation maxima during re-exposure after 1 week (long-term effects, timepoint 4).**

| k | $p_{FWE}$ | t | $p_{unc.}$ | x | y | z | label |
|---|---|---|---|---|---|---|---|
| **Main effect picture** | | | | | | | |
| *Neutral > Negative* | | | | | | | |
| No results | | | | | | | |
| *Negative > Neutral* | | | | | | | |
| 347 | <0.001 | 8.97 | <0.001 | 44 | -46 | -14 | Right Inferior Temporal Gyrus |
| 122 | <0.001 | 8.68 | <0.001 | -32 | -76 | 20 | Left Middle Occipital Gyrus |
| 85 | <0.001 | 8.01 | <0.001 | 18 | -6 | -14 | Right Hippocampus |
| 150 | 0.004 | 6.99 | <0.001 | -40 | -50 | -12 | Left Fusiform Gyrus |
| 84 | 0.005 | 6.90 | <0.001 | -30 | 22 | -20 | Left Inferior Frontal Gyrus |
| 79 | <0.001 | 5.23 | <0.001 | -24 | -4 | -14 | Left Amygdala (ROI) |
| 149 | <0.001 | 8.01 | <0.001 | 18 | -6 | -14 | Right Amygdala (ROI) |
| **Main effect regulation** | | | | | | | |
| *Distance > Permit* | | | | | | | |
| 23 | 0.005 | 4.28 | <0.001 | 24 | 0 | -22 | Right Amygdala (ROI) |
| *Permit > Distance* | | | | | | | |
| No results | | | | | | | |
| **Interaction effect picture X regulation** | | | | | | | |
| *(Distance Neutral > Permit Neutral) > (Distance Negative > Permit Negative)* | | | | | | | |
| No results | | | | | | | |
| *(Permit Neutral > Distance Neutral) > (Permit Negative > Distance Negative)* | | | | | | | |
| No results | | | | | | | |
| **Regulation effect negative pictures only** | | | | | | | |
| *DistanceNegative > PermitNegative* | | | | | | | |
| No results | | | | | | | |
| *PermitNegative > DistanceNegative* | | | | | | | |
| No results | | | | | | | |
| *Regulation effect neutral pictures only* | | | | | | | |
| *DistanceNeutral > PermitNeutral* | | | | | | | |
| 16 | 0.016 | 3.76 | <0.001 | 24 | -4 | -20 | Right Amygdala (ROI) |
| *PermitNeutral > DistanceNeutral* | | | | | | | |
| No results | | | | | | | |

Abbreviations: k = spatial extent, $p_{FWE}$ = *p*-values corrected for multiple comparisons (FWE), $p_{unc.}$ = uncorrected p-values, *t* = *t*-statistics, x, y, z = MNI coordinates. ROI indicates that an activation peak was observed within the left or right amygdala region of interest.

however, insular and cingulate regions were activated as well, indicating that these two regions specifically support more complex functions as compared to basic extinction processes, for which less distributed activation, primarily of the ventromedial prefrontal cortex, seems to be sufficient.

A third notable result is that a principled approach to experimental design and data analysis allows for further insights beyond the canonical results. On the one hand, as detailed below, we were able to identify different temporal response profiles for the sources and targets of emotional regulation. On the other hand, we were also able to differentiate between main and interaction effects for the picture and regulation conditions.

The first insight regards a dissociation between transient/sustained responses and the targets/sources of emotional regulation. During the stimulation phase, but not necessarily during the post-stimulation phase, we observed a primarily transient response pattern within the

amygdala, considering the statistical significance, spatial extent, and consistency across hemispheres. This pattern is consistent with its presumed alerting function [41]. In that sense, the amygdala initiates a response, but its activation does not represent the emotional response *per se*, which also includes down-stream activation in for example motor systems [42]. As long as the environment does not change any further, as it is the case for static pictures, continued activation of such an alerting structure is not needed, and hence the transient response. These results are compatible with Paret, Kluetsch, Ruf, Demirakca, Kalisch, Schmahl, et al. [43], who found that primarily the anterior insula, but also the left amygdala, exhibits a transient response pattern in event-related designs with longer picture durations. Both studies underscore that results crucially depend on the analytical model, and that modeling choices must be made consciously, based on a priori evidence.

**Concurrent emotion regulation effects in different picture conditions.** Another insight concerns the question whether or not emotion regulation is a domain-general or a domain-specific process; that is, whether regulation strategies and their effectiveness generalize across stimulus characteristics. Morawetz, Bode, Derntl and Heekeren [44] already showed, that activation of brain areas during emotion regulation generalizes across stimulus categories (IAPS pictures versus faces, film clips, reward, pain, scripts, and shapes). Our results point to a generalization across stimulus characteristics (in this case: valence), which is also in line with domain-general views on the brain basis of emotional reactivity [45]. The observation of decreased amygdala activation for neutral pictures during the distancing condition as compared to the permit condition may indicate that the distinction between negative and neutral pictures is a gradual one: we speculate that the mere presentation of (bright) pictures in the dark environment of the scanner elicits an unspecific alerting/arousing response–and that this response can be the target of emotional regulation.

Although, in our experiment, the difference between distance and permit was greater for negative than neutral pictures, the interaction effect between picture and regulation was relatively small, for example with regard to its spatial extent, and not as consistent as for the main effects. Similar results were also reported by Walter and colleagues [9] with a similar experimental design, indicating that there is a minor (quantitative), but no fundamental (qualitative) difference between distancing oneself from a negative or a neutral stimulus. The overall activation patterns of the neutral and negative pictures and the joint analysis are also similar in that they all lead to activation of the right middle frontal gyrus and inferior parietal lobe. This supports the idea of a core network consisting of middle frontal and inferior parietal regions involved in general regulation efforts, and that additional components are added to this network depending on the particular task at hand. From a conceptual point of view, we suggest that it is therefore not just either domain-general or just domain-specific processing that characterizes emotion regulation, but both—depending on the particular region.

## Effects of time

The second major focus of our study was on the temporal aspects of emotion regulation. We implemented a slow event-related design to achieve a balance between stimulation (timepoint 1) and post-stimulation (timepoint 2) periods, and also investigated the response to previously regulated and non-regulated pictures after short (10 min, timepoint 3) and long (one week, timepoint 4) intervals.

**Immediate emotion regulation aftereffects at timepoint 2.** We observed paradoxically increased amygdala activation after regulation at timepoint 2. However, these did not appear as a fully crossed interaction and were dependent on the statistical activation model: In the voxel-based analysis, we found greater activation during distance than during permit in the

post-stimulation phase only for sustained responses and interaction effects between time and regulation for both the transient and sustained responses. These results can be regarded as partial replications of the paradoxical rebound effect reported by Walter and colleagues [9]. They confirm the original proposition of post-stimulation activation differences across conditions as well as activation increases and decreases within conditions, but disagree with respect to the exact nature of this effect. For example, the original effect was found using the activation obtained from extracted time courses, whereas we used a model-driven approach. On the one hand, the effects depended on how the amygdala responses were modeled, and on the other hand, they did not necessarily occur at precisely the same locations (in terms of MNI coordinates), although all of them occurred within the amygdala ROI. Although this implies that a functional interpretation of this effect needs to be made in a more careful way, both studies support the notion of an interdependence of stimulation and post-stimulation periods. One could argue that the amount of cortical engagement during reappraisal is reduced as a function of time spent on implementing a certain reappraisal strategy [46]. This in turn might lead to an increase of amygdala activation after initial down-regulation.

Previous studies proposed that the concept of emotional aftereffects be extended to other regions apart from the amygdala, and to other types of interaction besides paradoxical, or crossed, ones. In this regard, a key result of Lamke and colleagues [22] is that reverse task-rest interactions (decreased activation during regulation but increased activation during relaxation/rest) were present in the amygdala after emotion down-regulation, whereas concordant task-rest interactions (increased activation during both regulation and relaxation/rest periods) were apparent in the prefrontal cortex. Comparable results were obtained in our study. We also observed task(stimulation)-rest(post-stimulation) interactions for negative pictures in regions beyond the amygdala, for example in the occipital cortex and the ventromedial frontal/subgenual cingulate cortex.

**Short- and long-term emotion regulation effects (timepoints 3 and 4).** In our study, previous regulation had also an impact on amygdala activation during the re-exposure sessions: both after 10 min and after 1 week: Previously down-regulated items showed greater activation within the left and right amygdala than previously non-regulated items. This effect was, however, not limited to negative pictures, but also appeared for neutral ones. These findings differ from previous results [9,24] indicating that emotion regulation effects persisted over time, that is that previously down-regulated items were associated with decreased amygdala activation. Hermann and colleagues [47] also investigated re-exposure to previously (one week) regulated stimuli and found not difference between previously distanced and permitted stimuli in amygdala activation nor in ratings of emotional experience. However, the authors reported an effect of previous reinterpretation, another reappraisal tactic, of negative stimuli leading to increased amygdala activation and reduced emotional experience during re-exposure. Additionally, there was no difference in re-exposure effects between reinterpretation and distancing. Since we did not record arousal ratings during re-exposure, we cannot conclude that being confronted with negative situations that have been previously regulated via distancing leads to reduced emotional experience. Nevertheless, the findings by Hermann and colleagues [47] might suggest, that this is a possible consequence of reappraisal, albeit at least for reinterpretation.

All of the aforementioned studies show that regulation effects persist over time, albeit in different ways. Although it has previously been shown that re-exposure effects also depend on the regulation strategy [48], this is no explanation for the specific pattern of results in our study, and why there are differences with regard to the Walter et al. study, given the similarity of the experimental setup. This remains an open question and needs to be the subject of further replication efforts.

## Limitations

We will now address the limitations of our study, which primarily concern the experimental design and the generality of our results. First, for reasons of design efficiency we only investigated negative vs. neutral pictures. This did not allow us to disentangle the impact of different levels of arousal and valence, since negative and neutral pictures simultaneously differ in both dimensions. It has been shown, though, that this is a relevant distinction with regard to amygdala [49] and prefrontal activation [50]. Including positive pictures would have allowed for a more comprehensive investigation in this regard. Further, a comparison of negative and neutral pictures alone does not allow conclusions about different kinds of emotion, if a discrete model of emotions is assumed. Again, there is evidence that different negative emotions such as fear, anxiety, sadness, and anger, and also positive vs. negative emotions are differentially susceptible to emotional regulation [51–53].

Second, this experiment was not designed to investigate the effects of different reappraisal tactics. While there is a common core network of emotion regulation, which we also found in our study, this does not necessarily mean that our results hold for different regulation tactics or even other strategies. Similar to the inclusion of further stimulus categories, the inclusion of further strategies would have been desirable, but would have necessitated the presentation of trials in a more rapid succession, which conflicted with our decision for a slow event-related experimental design. In that sense, there is a trade-off between design choices that cannot be resolved within a single study.

Third, the interpretation of our results rests on the assumption that participants on average could successfully implement the distance and permit instructions. Although we took effort to control for this by means of detailed instructions, training, and subsequent self-reports, it is possible that alternative strategies, involuntarily regulation, or mind-wandering were applied by the participants. We assume, though, that the impact of such regulation variants is random and just increases error variance in the data. However, to investigate whether such variability can lead to any systematic bias, further targeted studies are needed, e.g. to distinguish voluntary vs. involuntary emotion regulation.

Fourth, the interpretation of results for the re-exposure session is impacted by the drop-out of participants, which led to a reduced sample size for this part of the experiment. Also, additional arousal ratings during this phase could have provided additional information on the short- and long-term effects of emotion regulation. Finally, more variables than those considered for the present study play a role in emotion regulation: on the one hand, variation among individuals, be it clinical or non-clinical, will impact cognitive emotion regulation in a sense that different strategies will have different efficacies for different individuals. This could, for example, take the form of an association of successful reappraisal with less trait anxiety and more positive daily emotion as well greater activation in medial and lateral prefrontal regions [54]. On the other hand, successful vs. deficient emotion regulation will lead, in a developmental perspective, to different clinical trajectories; for example, emotion regulations skills predict the severity of anxiety symptoms across an interval of 5 years [55], and improving emotion regulation skills has been shown to enhance the efficacy of psychotherapeutic interventions [56]. Emotion regulation deficits are present in almost all clinical psychological disorders, which in turn are characterized by differential strategies of unsuccessful emotion regulation. Especially in the clinical context, comparing short term beneficial vs long term dysfunctional emotion regulation attempts appears very promising. To extend the study design to other populations than young and healthy students is therefore a necessary step for translating emotion regulation research into application contexts.

## Conclusions

Many of the issues central to this study have already been investigated, but not necessarily in a single study, and indeed, often in isolation. Further integration within the field requires some "firm ground", that is canonical results of established and replicable findings that can be derived from systematic reviews and quantitative meta-analyses, but also from replication efforts such as the present study. A key motivation of our study was to replicate and refine some core and emerging results in cognitive emotion regulation. Its main contribution is the joint and coherent consideration of a subset of relevant experimental factors–emotional processing and regulation as well as their temporal trajectories–and their effects on the activation patterns of known emotion generating and regulating networks. Our results confirm and extend previous characterizations of these networks, but also call for further investigation especially of their temporal dynamics.

## Supporting information

**S1 Fig. Overview over the larger project and corresponding peer-reviewed publications and manuscripts, respectively.**
(DOCX)

## Acknowledgments

We thank Patricia Schimm for her great support with providing Open Data and Open Materials.

## Author Contributions

**Conceptualization:** Kersten Diers, Denise Dörfel, Sabine Schönfeld, Henrik Walter, Alexander Strobel, Burkhard Brocke.

**Data curation:** Kersten Diers, Anne Gärtner.

**Formal analysis:** Kersten Diers.

**Funding acquisition:** Sabine Schönfeld, Alexander Strobel, Burkhard Brocke.

**Investigation:** Kersten Diers.

**Methodology:** Kersten Diers, Denise Dörfel, Anne Gärtner, Sabine Schönfeld, Henrik Walter, Alexander Strobel, Burkhard Brocke.

**Project administration:** Kersten Diers, Alexander Strobel.

**Resources:** Alexander Strobel, Burkhard Brocke.

**Supervision:** Denise Dörfel, Henrik Walter, Alexander Strobel, Burkhard Brocke.

**Visualization:** Kersten Diers, Denise Dörfel.

**Writing – original draft:** Kersten Diers.

**Writing – review & editing:** Kersten Diers, Denise Dörfel, Anne Gärtner, Sabine Schönfeld, Henrik Walter, Alexander Strobel, Burkhard Brocke.

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
