## [Decision Letter · Decision Letter 0]

12 Feb 2021

PONE-D-20-39022

Should we detach from detachment? Regulatory and post-regulatory effects of emotion downregulation.

PLOS ONE

Dear Dr. Dörfel,

Thank you for submitting your manuscript to PLOS ONE. After careful consideration, we feel that it has merit but does not fully meet PLOS ONE’s publication criteria as it currently stands. Therefore, we invite you to submit a revised version of the manuscript that addresses the points raised during the review process.

Congratulations on your interesting contribution. Both expert reviewers have commented positively on your manuscript, but they have also made several constructive recommendations to make your manuscript more convincing and stronger. Please very carefully attend to each of the reviewers' points and please make sure to also integrate required additional information in the manuscript, not only in the response letter to the reviewers.

We look forward to receiving your revised manuscript.

Kind regards,

Ilona Papousek

Academic Editor

PLOS ONE

Journal Requirements:

2. Please note that PLOS ONE does not copy edit accepted manuscripts (https://journals.plos.org/plosone/s/criteria-for-publication#loc-5). To that effect, please ensure that your submission is free of typos and grammatical errors.

3. We noted in your submission details that a portion of your manuscript may have been presented or published elsewhere.

"No.

Results from the present sample on the research questions of this manuscript have not been reported previously nor are under consideration for publication elsewhere.

However, the presented study is part of a larger project of which several reports have already been published (Diers et al., 2014; Gärtner et al., 2019; Scheffel et al., 2019; Dörfel et al., 2020). Diers et al. (2014) reported results from a different sample on a different research question of the project, Gärtner et al. (2019), Scheffel et al. (2019), and Dörfel et al. (2020) combined data of this manuscript's sample with a third sample, but analyzed completely different research questions, each."

Please clarify whether this publication was peer-reviewed and formally published. If this work was previously peer-reviewed and published, in the cover letter please provide the reason that this work does not constitute dual publication and should be included in the current manuscript.

Reviewers' comments:

Reviewer's Responses to Questions

**Comments to the Author**

1. Is the manuscript technically sound, and do the data support the conclusions?

Reviewer #1: Yes

Reviewer #2: Partly

2. Has the statistical analysis been performed appropriately and rigorously? 

Reviewer #1: Yes

Reviewer #2: Yes

3. Have the authors made all data underlying the findings in their manuscript fully available?

Reviewer #1: Yes

Reviewer #2: No

4. Is the manuscript presented in an intelligible fashion and written in standard English?

Reviewer #1: Yes

Reviewer #2: Yes

5. Review Comments to the Author

Reviewer #1: In this paper, the authors present a complex, but interesting study in which they compare neural correlates of regulating emotions via detachment vs. non-regulation in a sample of 46 participants. Brain activation effects of regulating vs permitting negative and neutral affective pictures are compared at four different time points: 1) during regulation, 2) immediately post-regulation, 3) at a short re-exposure after 10 min, and 4) at a long re-exposure after 1 week. For the main findings, the authors report that detachment (compared to non-regulation) led to a decreased amygdala response but increased frontal and parietal response during regulation. Interestingly however, this response was reversed immediately post-regulation, with a higher amygdala response reported for detachment than for non-regulation. This “paradoxical” effect was maintained for the short and long re-exposure intervals, with higher amygdala activation observed for pictures participants had previously detached from. In general, the study seems well-executed, the reported analyses are sound, and the provided figures help in understanding the complexity of the findings.

However, a few issues remain. Below, I address some points that may hopefully clarify and improve the manuscript.

##Abstract

# 1 The denoted, recruited sample size of N = 48 does not represent the final sample size. In the methods, the authors specify that MRI data of n = 46 participants was available for timepoints 1 to 3 (regulation, post-regulation, short re-exposure) and that data of n = 30 participants was available for timepoint 4 (re-exposure after 1 week). This should be clarified in the abstract.

# 2 Since there are a lot of different findings in this study, the main findings should be reported in a more clear and comprehensible way, always mentioning both tasks (detachment vs. non-regulation). I also suggest adding timepoints (or equivalent organizers) to the reported conditions. For example: “During regulation (timepoint 1), amygdala activation was lower during detachment that during non-regulation […]. During the post-regulation interval (timepoint 2), however, this effect was reversed, indicating greater amygdala activation after detachment that after non-regulation […].

# 3 When reporting the main findings, substitute “cognitive emotion regulation” with “detachment” to improve clarity.

# 4 I spotted two minor grammatical errors:

In the sentence starting with “Similarly, after 10 minutes […]”, the comma after “pictures” should be removed.

In the second to last sentence, “across brain region” should be “across brain regions”.

##Introduction

# 1 Major point: The introduction is heavily focused on brain activation during cognitive emotion regulation in general, which is fine. However, what is missing is a framework that introduces detachment as an emotion regulation strategy, which the authors introduce as part of the cognitive reappraisal family of emotion regulation strategies. Yet, this only becomes evident in the discussion, where detachment is frequently addressed as a reappraisal strategy, yet without any previous explanations in the introduction, this is rather confusing. Accordingly, the introduction needs to be revised to include a) a clear definition of detachment, b) framing detachment as a cognitive reappraisal strategy, and especially c) explanations why detachment in particular was chosen for the present study (often used by individuals? Effective for down-regulation of negative emotions as reported by previous literature?)

# 2 In paragraph 1, the last sentence is very difficult to read/understand: “At a neural level, this likely implemented by the action of emotion-regulating regions in the cortex upon typically subcortical emotion-generating regions”. I suggest rephrasing is as “At a neural level, emotion regulation is likely implemented by cortical regions that exercise influence over emotion-generating subcortical regions”.

# 3 Paragraph 2, last sentence: It should be clarified that the “indispensable control network” is an indispensable control network of cognitive emotion regulation.

#4 Page 4, paragraph 1, last sentence starting with “This concept has been extended […]. This is very difficult to read due to its length and should be divided up.

# 5 I am confused by the paragraph starting with “The study of Walter, von Kalckreuth […]” which ends with “[…] was negatively correlated to the paradoxical aftereffect”. What is the paradoxical aftereffect here? What is it negatively correlated with? Am I right in the assumption that immediately after regulation, there was amygdala upregulation, but minutes later, there was amygdala downregulation? Please elaborate on this and specify that there were two distinct post-regulation phases.

# 6 Page 5, paragraph 1, last sentence. “Another example of long-term effects been established […]” should be corrected to “Another example of long-term effects HAS been established […]”

# 7 Page 5, paragraph 2. The authors state that “systematic investigations of the temporal dimension of emotion regulation are still lacking”, however, in the paragraphs before, they elaborated on quite a few temporal investigations on emotion regulation, which is somewhat confusing. This sentence should either be specified to “systematic investigations of the temporal dimension of detachment” or rephrased to simply reflect the “need for more research” or “need for replication”.

# 8 Page 6, paragraph 1, sentence 2. There is a typo. “shouldbe” instead of “should be”

# 9 Page 6, paragraph 1, last sentence. This expectation would benefit from referencing previous studies again that showed similar effects (Lamke et al., 2016; Walter et al., 2015)

##Methods

# 1 Materials and methods: I am confused about the sentence “We report how we determined our sample size […] in the study [22]”. Which study does this refer to? Does this directly relate to reference source nr. 22? If yes, this source is not accessible to me and may not be to other readers. Therefore, I suggest to briefly elaborate on sample size determination and data exclusion in the participants section of this manuscript.

# 2 Emotion regulation task: When reading the author’s instructions for the non-regulation condition, i.e., “not to voluntary intensify emotions, not re-interpret the situation, no distraction”, I wondered whether participants’ engagement of some other type of emotion regulation could potentially explain the paradoxical amygdala effects in later time-points. I realize that the authors took great effort with training sessions to ensure that participants knew how to comply with the paradigm, but perhaps there was some involuntary acceptance, or attentional deployment, nonetheless? I would also consider mind-wandering, since participants could have felt that the “permit” instruction meant they did not have to pay close attention to the task? I recommend that the possibility of covert/automatic regulation processes during the permit condition should be briefly addressed in the study limitations.

# 3 Emotion regulation task: Please specify the range of the continuous arousal scale in this study (1 to 100?)

# 4 Re-exposure task: Please elaborate why the exact presentation time of 1000 ms was chosen for re-exposure. Is this choice based on previous studies that substantiate 1000 ms is sufficient/recommended to measure emotional reactivity?

# 5 Stimuli: In the first sentence, two references have not been converted to the reference style of Plos One and are also missing from the reference list.

##Results

# 1 Behavioral analysis: For clarity, mean values and standard deviations should be added to the behavioral results of participants’ subjective arousal ratings.

# 2 I recommend adding subheadings to the results section. For 3.2.1, “activation differences between neutral and negative pictures”, “activation differences between detachment and non-regulation”, and “interaction effects between valence and strategy” would help. Similarly, for 3.2.5., subheadings for “re-exposure after 10 min” and “re-exposure after 1 week” should be added.

##Discussion

# 1 Major point: The discussion is extremely detailed, long (over 10 pages) and thus, hard to read. Certain parts should be shortened in order to improve readability and comprehension. I realize that due to the multitude of time points and respective findings, this is not an easy task. More sub-headings may help organize the amount of information in a better way and may allow for more structured reading and fewer repetitions of findings throughout the discussion. I would also suggest shortening the paragraph on “domain-general” and “domain-specific” effects of emotion regulation. Additionally, perhaps parts of the explanations on reappraisal findings can be moved to the introduction (provided that detachment is framed properly as a reappraisal strategy in the introduction). I also noted lengthy in-text references to author names, which are better abbreviated in square brackets.

# 2 For the beginning of the discussion and the summary of findings for the different time points, I also suggest a numbering system and more indicative phrasing, e.g., “During the regulation (timepoint 1), it was found that…”, “Immediately after (timepoint 2), amygdala activation….” (see my suggestions for the abstract).

# 3 I suggest rephrasing certain sentences to improve readability.

- Page 25, lines 169 to 171, starting with “we were able to identify”. This should be split into two sentences for better comprehension.

- Page 29, lines 249 ff. I suggest rephrasing the sentence as “Once could argue that the amount of cortical engagement during reappraisal is reduced as a function of time spent on implementing a certain reappraisal strategy.”

- Page 29, lines 252 to 253. I am not sure I understand this sentence correctly, but I think it should read “Previous studies proposed that the concept of emotional aftereffects be extended to other regions apart from the amygdala […]”.

# 4 I spotted some minor grammatical errors/typos:

Page 24, first sentence, line 138. “during detachment of negative pictures” should be “during detachment from negative pictures” or alternatively “during the application of detachment to negative pictures”.

Page 25, first paragraph, line 164. “activated” should be “were activated”.

Page 27, second paragraph, lines 216-217. The verb “supports” appears twice in the sentence, and it is hard to tell what the meaning is. Perhaps this was meant as “This supports the idea of a core network consisting of […] regions INVOLVED in general regulation efforts […]” ?

Page 30, line 279, “[…] depend on regulation strategy” should be “[…] depend on THE regulation strategy.”

# 5 I am confused by the concluding sentence (p. 32, lines 335 to 337). What do the authors mean by “reducing the fragmentation” in the field of cognitive emotion regulation research? The meaning behind this statement should be explained.

Reviewer #2: This fMRI study investigated the immediate and post-regulatory effects of emotion regulation via detachment in a sample of 48 healthy participants. Besides the emotion regulation phase, effects of emotion regulation were investigated for the post-regulation phase (directly after each trial), for re-exposure after 10 minutes and re-exposure after one week. The results especially show a time-dependent up- and down-regulation of amygdala activation in response to differing picture categories (negative/neutral/both) for the different phases. This is a highly relevant research question. In principle, the manuscript is well-written and the study procedure methodologically sound. However, the authors do not consequently distinguish between regulation of emotions (in response to negative pictures), detachment during neutral picture viewing or detachment during both picture types. Together with different analysis approaches for amygdala activation and varying results for the different picture categories, it is hard to understand the main message of the manuscript. I hope that the following comments might help to improve the manuscript.

Introduction

1. The authors did not really differentiate between different reappraisal tactics even though there is evidence from previous studies regarding their differential effects (e.g. Dörfel et al., 2014; Hermann et al. 2020). I would appreciate the authors to elaborate on this issue.

2. Page 5: Another example of long-term effects been established for episodic memory processes: -> incomplete sentence

3. This shouldbe accompanied by (page 6) -> should be

Methods and materials

4. The authors state that all data and material files are available from the Open Science Framework. I could however not find data from preprocessing or first-level models online. The authors should indicate which data are not provided and why, and/or add the respective data.

5. How was the trial order during the emotion regulation task? This is not explicitly stated in the manuscript.

6. Another issue concerns an aspect of the instruction during the re-exposure phase (“Specifically, they were instructed not to voluntarily change their emotional experience as they had done during the main experiment.”). Why did the authors decide to prevent the participants from spontaneously applying the prior regulation strategy during re-exposure (as a potential after-effect of emotion regulation)?

7. The authors used three different types of strategies for analyzing amygdala activation. The authors should explicitly state that they did two different first-level analyses for the amygdala (transient/sustained response). This is not entirely clear for me.

8. Did the authors use a high-pass filter?

9. How were the masks for the amygdala created in detail? Which probability threshold was used?

10. The contrasts are not described in the methods section.

Results

11. It is very time-consuming to understand the amygdala results which are spread over the main manuscript and the supplement. I suggest to integrate the main results (post-regulation, re-exposure) for all contrasts (at least for the amygdala) into the main manuscript. Moreover, it should explicitly be stated from which analyses the amygdala results stem (in Tables, Figures, main manuscript, supplement).

Discussion

12. Throughout the abstract and discussion, it is not entirely clear for me which interpretation is related to which findings, especially regarding amygdala activation. If I understand it right, the immediate and short-/long-term regulation effects were partly found for negative pictures, neutral pictures or both picture categories (main effect). It is a very interesting result that amygdala activation is also decreased during detachment regarding neutral pictures. However, I do not understand why the authors interpret this effect as emotion regulation, as there are by definition no emotions elicited during neutral picture viewing. I would request the authors to elaborate on this issue. Moreover, it should be clearly stated throughout the manuscript (e.g. abstract, discussion) for which picture categories the effects have been found and what this means with regard to ‘emotion’ regulation or detachment irrespective of emotional content.

13. Moreover, the authors did not mention the result of increased right amygdala activation during detachment for negative compared with neutral pictures (emotion regulation phase) in the results section nor discussed this finding (see Table 1, last line). As enhanced amygdala activation for detachment > permit is also found for the short- (negative pictures) and long-term (main effect strategy; neutral pictures) re-exposure phase, I can’t understand why enhanced amygdala activation is only discussed for the re-exposure phases. How do the authors interpret this up-regulation of amygdala activation (sustained response) during detachment (emotion regulation phase) also in relation to the re-exposure findings?

14. The authors discuss the primarily transient response pattern within the amygdala (page 25). I do not understand on which results this interpretation is based , as there are both amygdala results for ROI and ROI/stick model.

15. The authors might discuss further differences to other studies/limitations of their study: the smaller sample size for re-exposure after one week, sex difference, lacking ratings during re-exposure,…

6. PLOS authors have the option to publish the peer review history of their article (what does this mean?). If published, this will include your full peer review and any attached files.

Reviewer #1: No

Reviewer #2: No

---

## [Author Response · Author response to Decision Letter 0]

11 May 2021

Changes to the manuscript that were not requested by the reviewers:

We have introduced the following changes that were not requested by the reviewers: 

First, we have detected a mistake in the final paragraph of the introduction, i.e. the hypotheses section. In the submitted version, this was “We expected that the lower amygdala activation in the detach condition as compared to the permit condition in the regulation stimulation phase reverses in the post-regulation phase (immediate aftereffect) as well as during re-exposure”. In the revised version, this sentence is split into two sentences: “We expected that the lower amygdala activation in the distance condition as compared to the permit condition in the stimulation phase reverses in the post-stimulation phase (immediate aftereffect, timepoint 2). We further aimed at characterizing the amygdala response during re-exposure with the same pictures after short (10 minutes, timepoint 3) and long (1 week, timepoint 4) time intervals". The difference between the old and new variant is in the expected direction of the re-exposure effects: the originally submitted sentence contained an error that resulted from the erroneous contraction of two sentences into one during an internal revision of the manuscript. This error had been overlooked despite its misleading implications. The revised form is again consistent with the original project proposal. It is now also consistent with the discussion and interpretation of the re-exposure effects that had been part of the original submission. We apologize for the lack of attention on our part that led to the error. We certainly see that post-hoc changes in hypotheses are not acceptable, but would like to stress that this is not the case here. In particular, we are not changing the hypotheses into the direction of the results, and what we expected in the first place (and should have been mentioned in the introduction) can also be inferred from the discussion which is left unchanged in this regard.

Second, we have removed Supplementary Figure 1, which displayed a graph of the behavioral arousal ratings. This was the only remaining content of the supplementary materials, after we included the supplementary tables into the main manuscript, as suggested by the reviewers. We also included the Mean and SD values of the behavioral ratings in the manuscript, and therefore we felt that the figure was redundant and hence removed it. As a consequence, a supplement is no longer needed.

Reviewer #1: 

Comment: In this paper, the authors present a complex, but interesting study in which they compare neural correlates of regulating emotions via detachment vs. non-regulation in a sample of 46 participants. Brain activation effects of regulating vs permitting negative and neutral affective pictures are compared at four different time points: 1) during regulation, 2) immediately post-regulation, 3) at a short re-exposure after 10 min, and 4) at a long re-exposure after 1 week. For the main findings, the authors report that detachment (compared to non-regulation) led to a decreased amygdala response but increased frontal and parietal response during regulation. Interestingly however, this response was reversed immediately post-regulation, with a higher amygdala response reported for detachment than for non-regulation. This “paradoxical” effect was maintained for the short and long re-exposure intervals, with higher amygdala activation observed for pictures participants had previously detached from. In general, the study seems well-executed, the reported analyses are sound, and the provided figures help in understanding the complexity of the findings.

However, a few issues remain. Below, I address some points that may hopefully clarify and improve the manuscript.

Response: 

We thank the reviewer for their thorough, helpful and constructive comments, which are of great value for improving the manuscript. Below, we reply to the individual comments. Please note that, wherever we give references to pages or line numbers, these refer to the revised manuscript without track changes enabled.

Comment: 

##Abstract

# 1 The denoted, recruited sample size of N = 48 does not represent the final sample size. In the methods, the authors specify that MRI data of n = 46 participants was available for timepoints 1 to 3 (regulation, post-regulation, short re-exposure) and that data of n = 30 participants was available for timepoint 4 (re-exposure after 1 week). This should be clarified in the abstract.

Response: We have adjusted the numbers to the final sample size in the abstract.

Comment: 

# 2 Since there are a lot of different findings in this study, the main findings should be reported in a more clear and comprehensible way, always mentioning both tasks (detachment vs. non-regulation). I also suggest adding timepoints (or equivalent organizers) to the reported conditions. For example: “During regulation (timepoint 1), amygdala activation was lower during detachment that during non-regulation […]. During the post-regulation interval (timepoint 2), however, this effect was reversed, indicating greater amygdala activation after detachment that after non-regulation […].

Response: We very much agree and have improved the abstract (and also the methods, results and discussion section accordingly) as follows: In the introductory part of the abstract, we more clearly labeled effects during active emotion regulation as ‘concurrent’ (see also in the introduction) and effects at later time intervals as immediate, short-term and long-term, respectively. 

We renamed the active regulation phase (between picture onset and offset) as ‘stimulation phase’ and furthermore have marked this phase more clearly as ‘timepoint 1’, we renamed the post-regulation phase as ‘post-stimulation phase’ (after picture offset) and added ‘timepoint 2’. The re-exposure tasks/phases were renamed as timepoint 3 (10 minutes later) and timepoint 4 (one week later), respectively. This structure is also implemented in the methods, results and discussion section. Following this we had to shorten the abstract again which we achieved by rephrasing some sentences. The results-part of the abstract now reads:

“At timepoint 1, negative pictures (versus neutral pictures) elicited a strong response in regions of affective processing, including the amygdala. Distancing (as compared to permit) led to a decrease of this response, and to an increase of activation in the right middle frontal and inferior parietal cortex. We observed an interaction effect of time (stimulation vs. post-stimulation) and regulation (distance vs. permit), indicating a partial reversal of regulation effects during the post-stimulation phase (timepoint 2). Similarly, after 10 minutes (timepoint 3) and after 1 week (timepoint 4), activation in the amygdala was higher during pictures that participants were previously instructed to distance from as compared to permit. “

Comment:

# 3 When reporting the main findings, substitute “cognitive emotion regulation” with “detachment” to improve clarity.

Response: We corrected the sentence accordingly. Please note, that we also renamed detachment as distancing, this is related to one of your following comments. 

Comment:

# 4 I spotted two minor grammatical errors:

In the sentence starting with “Similarly, after 10 minutes […]”, the comma after “pictures” should be removed. In the second to last sentence, “across brain region” should be “across brain regions”.

Response: We corrected these errors as requested.

##Introduction

Comment:

# 1 Major point: The introduction is heavily focused on brain activation during cognitive emotion regulation in general, which is fine. However, what is missing is a framework that introduces detachment as an emotion regulation strategy, which the authors introduce as part of the cognitive reappraisal family of emotion regulation strategies. Yet, this only becomes evident in the discussion, where detachment is frequently addressed as a reappraisal strategy, yet without any previous explanations in the introduction, this is rather confusing. Accordingly, the introduction needs to be revised to include a) a clear definition of detachment, b) framing detachment as a cognitive reappraisal strategy, and especially c) explanations why detachment in particular was chosen for the present study (often used by individuals? Effective for down-regulation of negative emotions as reported by previous literature?)

Response: 

Thank you for this very important comment. We included the framework of {Powers, 2019, 30502352@@author-year} into the introduction in order to offer a definition, description and classification of detachment/distancing according to and extending the Process Model by {Gross, 1998, 9457784@@author-year}. Of particular relevance for our manuscript is the distinction between “reinterpretation” and “distancing” as specific tactics of the ER strategy “reappraisal” (cognitive change). As a consequence, we replaced the term “detachment” with “distancing” in order to adapt our terminology to the framework. We also added explanations why distancing in particular was chosen for the present study. We also shortly describe the different forms of distancing and explain, which form we used in our investigation and why. This is just a change in wording, with no further implications for the results or their interpretation. We believe, however, that this is a step towards unified terminology and helps in comparing and aggregating results across studies. 

You will find the new paragraphs at page 3, line 45 ff.

Comment:

# 2 In paragraph 1, the last sentence is very difficult to read/understand: “At a neural level, this likely implemented by the action of emotion-regulating regions in the cortex upon typically subcortical emotion-generating regions”. I suggest rephrasing is as “At a neural level, emotion regulation is likely implemented by cortical regions that exercise influence over emotion-generating subcortical regions”.

Response: Due to the new paragraphs in the introduction at pages 3 and 4, the respective sentence has moved to page 4 (line 77). We rephrased the sentence accordingly. 

It now reads: 

“At a neural level, emotion regulation, including cognitive change and attentional deployment, is implemented by cortical regions that exercise influence over emotion-generating subcortical regions [13].

Comment: 

# 3 Paragraph 2, last sentence: It should be clarified that the “indispensable control network” is an indispensable control network of cognitive emotion regulation.

Response: We corrected the sentence accordingly. It now reads: “Taken together, this suggests that while a common and potentially indispensable control network of cognitive emotion regulation can be identified, additional structures are selectively recruited only under certain circumstances, leading to highly context-dependent patterns of co-activation of different brain structures. “

Comment: 

# 4 Page 4, paragraph 1, last sentence starting with “This concept has been extended […]. This is very difficult to read due to its length and should be divided up.

Response: We rephrased the sentence accordingly. It now reads: “This concept has been extended to a number of brain regions and timepoints. Lamke and colleagues [22] observed decreased activation during emotion regulation in the amygdala and in visual cortex regions in the stimulation phase but increased activation in the subsequent post-stimulation phase.”

Comment: 

# 5 I am confused by the paragraph starting with “The study of Walter, von Kalckreuth […]” which ends with “[…] was negatively correlated to the paradoxical aftereffect”. What is the paradoxical aftereffect here? What is it negatively correlated with? Am I right in the assumption that immediately after regulation, there was amygdala upregulation, but minutes later, there was amygdala downregulation? Please elaborate on this and specify that there were two distinct post-regulation phases.

Response: Sorry for the confusion. With the paradoxical aftereffect we refer to increased amygdala activation for previously down-regulated pictures that was observed during the immediate post-stimulation phase (immediately after picture offset, hence after active emotion regulation). It was negatively correlated with amygdala activation during a passive-viewing re-exposure task. In that sense, there were stimulation (with active emotion regulation vs. no-regulation) and post-stimulation phases during the main experiment, and a subsequent re-exposure after several minutes. We have clarified this experimental design in the manuscript. We also hope that by changing the wording in the introduction up to this paragraph, it becomes clearer. 

The paragraph now reads: “One study [9] demonstrated not only immediate aftereffects, but also effects across an even more extended period of time up to several minutes (short-term effects). Specifically, upon re-encountering previously regulated negative stimuli in a passive viewing task ten minutes later, the authors observed a persisting down-regulation of the amygdala. This re-exposure effect was negatively correlated to the paradoxical aftereffect that was observed immediately after the stimulation phase; i.e., a higher immediate aftereffect was associated with a smaller short-term regulation effect.”

Comment: 

# 6 Page 5, paragraph 1, last sentence. “Another example of long-term effects been established […]” should be corrected to “Another example of long-term effects HAS been established […]”

Response: We corrected the sentence accordingly.

Comment: 

# 7 Page 5, paragraph 2. The authors state that “systematic investigations of the temporal dimension of emotion regulation are still lacking”, however, in the paragraphs before, they elaborated on quite a few temporal investigations on emotion regulation, which is somewhat confusing. This sentence should either be specified to “systematic investigations of the temporal dimension of detachment” or rephrased to simply reflect the “need for more research” or “need for replication”. 

Response: We think, that systematic approaches including different time intervals are not existent. We rephrased the sentence to make this point clearer. It now reads “As systematic investigations of the temporal dimension of distancing including several time intervals are still lacking, the present research aims at replicating and extending previous results on the temporal dynamics of distancing, that is its concurrent, immediate, short- and long-term consequences.”

Comment: # 8 Page 6, paragraph 1, sentence 2. There is a typo. “shouldbe” instead of “should be”

Response: We corrected the sentence accordingly.

Comment: 

# 9 Page 6, paragraph 1, last sentence. This expectation would benefit from referencing previous studies again that showed similar effects (Lamke et al., 2016; Walter et al., 2015)

Response: We added Lamke et al. (2014) and Walter et al. (2009) which were very similar to our study and served as starting point for our study design.

##Methods

Comment: 

# 1 Materials and methods: I am confused about the sentence “We report how we determined our sample size […] in the study [22]”. Which study does this refer to? Does this directly relate to reference source nr. 22? If yes, this source is not accessible to me and may not be to other readers. Therefore, I suggest to briefly elaborate on sample size determination and data exclusion in the participants’ section of this manuscript.

Response: Apologies for the misunderstanding. “Study” refers to our study, and the citation is different from that. The cited source refers to a recommendation about reporting participant and sample characteristics in order to maximize transparency. We have re-worded this sentence. It now reads: 

“We report how we determined our sample size, all data exclusions (if any), all manipulations, and all measures in our study, as recommended by transparency guidelines [28].”

We also added details about sample size determination to the first paragraph of the methods section (“Participants” section, page 9). Given that there are multiple outcomes for the current study, we did not calculate the sample size for a specific effect, but wanted to ensure that we are able to detect effects of a certain magnitude with a sample size that we expected to be able to acquire. Details about data non-availability / participant exclusion were already included in the same paragraph and were therefore left unchanged.

Comment: 

# 2 Emotion regulation task: When reading the author’s instructions for the non-regulation condition, i.e., “not to voluntary intensify emotions, not re-interpret the situation, no distraction”, I wondered whether participants’ engagement of some other type of emotion regulation could potentially explain the paradoxical amygdala effects in later time-points. I realize that the authors took great effort with training sessions to ensure that participants knew how to comply with the paradigm, but perhaps there was some involuntary acceptance, or attentional deployment, nonetheless? I would also consider mind-wandering, since participants could have felt that the “permit” instruction meant they did not have to pay close attention to the task? I recommend that the possibility of covert/automatic regulation processes during the permit condition should be briefly addressed in the study limitations.

Response: Yes, we agree that this is a good point and should be acknowledged as a caveat regarding the interpretation of the results. We have added a short paragraph to the limitations section (see page 37, last but one paragraph). We suggest that while such alternative forms of emotion regulation may have been present in our experiment, it is not likely that there is a systematic bias, but rather a random influence that increases the heterogeneity of observed effects, i.e. the error variance in the data.

Comment: 

# 3 Emotion regulation task: Please specify the range of the continuous arousal scale in this study (1 to 100?)

Response: The scale for the continuous arousal scale ranged from –200 to 200 for technical reasons (screen coordinates). However, the particular choice of scale units and limits did not affect the participants’ ratings, since only verbal anchors were provided at either end of the scale (“not at all aroused” vs. “very much aroused”), and participants had to move a slider to a position between those anchors that resembled their arousal. In other words, no numerical information was given to participants. We added this clarification to the methods section (page 11, end of chapter “Emotion regulation task (timepoints 1 and 2)”).

Comment: 

# 4 Re-exposure task: Please elaborate why the exact presentation time of 1000 ms was chosen for re-exposure. Is this choice based on previous studies that substantiate 1000 ms is sufficient/recommended to measure emotional reactivity?

Response: The choice of 1000 ms presentation duration in the emotional reactivity task is based on the Walter et al. (2009) study. In this study, the same setting was used for a reactivity task that apparently succeeded in re-eliciting emotion-related amygdala activation. We aimed at maximum comparability with this study, and so chose the same task duration. A reference to this study has been added to the manuscript (page 11, paragraph “Re-exposure task (timepoints 3 and 4)”).

Comment: 

# 5 Stimuli: In the first sentence, two references have not been converted to the reference style of Plos One and are also missing from the reference list.

Response: We apologize for the mistake, corrected the reference style and added the references to the list.

##Results

Comment: 

# 1 Behavioral analysis: For clarity, mean values and standard deviations should be added to the behavioral results of participants’ subjective arousal ratings.

Response: The mean and SD values have been added to the manuscript. Please note, that the accompanying figure, originally contained in the supplementary material, has been removed, because all other supplementary tables were included in the main document after a recommendation of reviewer #2. 

Comment: 

# 2 I recommend adding subheadings to the results section. For 3.2.1, “activation differences between neutral and negative pictures”, “activation differences between detachment and non-regulation”, and “interaction effects between valence and strategy” would help. Similarly, for 3.2.5., subheadings for “re-exposure after 10 min” and “re-exposure after 1 week” should be added.

Response: Since PLOS does not allow for headings at the fourth level, we rephrased all headings of the results section to provide more structure and to implement your recommendation. The headings in the results section now are:

“Behavioral analysis of the stimulation phase (concurrent effects, timepoint 1)”

“Neuronal activation differences during the stimulation phase (concurrent effects, timepoint 1)”

 “Activation differences between negative and neutral pictures”

 “Activation differences between distancing and permitting emotions”

 “Interaction effects between picture and regulation”

“Neuronal interaction effects of regulation (permit, distance) and time (stimulation, post-stimulation)”

“Neuronal interaction effects of regulation and time (immediate aftereffects, timepoint 2)”

“Activation time-courses during the stimulation- and post-stimulation phase (timepoint 1, timepoint 2)” 

“Neuronal activation differences during the re-exposure tasks” 

 “Re-exposure after 10 minutes (short-term effects, timepoint 3)”

 “Re-exposure after 1 week (long-term effects, timepoint 4)” 

##Discussion

Comment: 

# 1 Major point: The discussion is extremely detailed, long (over 10 pages) and thus, hard to read. Certain parts should be shortened in order to improve readability and comprehension. I realize that due to the multitude of time points and respective findings, this is not an easy task. More sub-headings may help organize the amount of information in a better way and may allow for more structured reading and fewer repetitions of findings throughout the discussion. I would also suggest shortening the paragraph on “domain-general” and “domain-specific” effects of emotion regulation. Additionally, perhaps parts of the explanations on reappraisal findings can be moved to the introduction (provided that detachment is framed properly as a reappraisal strategy in the introduction). I also noted lengthy in-text references to author names, which are better abbreviated in square brackets.

Response: Yes, we partly agree and were also aware of the rather long discussion. We now shortened several paragraphs (including the one on “domain-general” and “domain-specific” effects) and added additional sub-headings to increase readability. However, due to recommendations from both reviewers we also added some points to the discussion (for instance under ‘limitations’). The lengthy in-text references were due to the PLOS reference style, we changed the parts manually. Overall, we consider the discussion section much more understandable and concise now. 

Comment:

# 2 For the beginning of the discussion and the summary of findings for the different time points, I also suggest a numbering system and more indicative phrasing, e.g., “During the regulation (timepoint 1), it was found that…”, “Immediately after (timepoint 2), amygdala activation….” (see my suggestions for the abstract).

Response: Thank you very much for this great suggestion, we implemented this numbering system throughout the whole manuscript. We also added a more concise description of the different timepoints and intervals in the introduction and continued this in the methods and results section. Concurrent effects during active regulation are now consequently labeled as timepoint 1 (stimulation phase of the ER experiment), immediate (after)effects after picture offset are labeled as timepoint 2 (post-stimulation phase), short-term effects after 10 minutes (re-exposure task session 1) as timepoint 3, and long-term effects after 1 week (re-exposure task session 2) as timepoint 4. We also renamed and added subheadings, respectively, in the methods, results and discussion sections and hope the temporal dynamics of our study and therefore of ER effects become clearer now. 

Comment: 

# 3 I suggest rephrasing certain sentences to improve readability.

- Page 25, lines 169 to 171, starting with “we were able to identify”. This should be split into two sentences for better comprehension.

- Page 29, lines 249 ff. I suggest rephrasing the sentence as “Once could argue that the amount of cortical engagement during reappraisal is reduced as a function of time spent on implementing a certain reappraisal strategy.”

- Page 29, lines 252 to 253. I am not sure I understand this sentence correctly, but I think it should read “Previous studies proposed that the concept of emotional aftereffects be extended to other regions apart from the amygdala […]”.

Response: Thank you for these suggestions; they indeed improve readability, and we have adjusted the sentences accordingly. 

Comment:

# 4 I spotted some minor grammatical errors/typos:

Page 24, first sentence, line 138. “during detachment of negative pictures” should be “during detachment from negative pictures” or alternatively “during the application of detachment to negative pictures”.

Page 25, first paragraph, line 164. “activated” should be “were activated”.

Page 27, second paragraph, lines 216-217. The verb “supports” appears twice in the sentence, and it is hard to tell what the meaning is. Perhaps this was meant as “This supports the idea of a core network consisting of […] regions INVOLVED in general regulation efforts […]” ?

Page 30, line 279, “[…] depend on regulation strategy” should be “[…] depend on THE regulation strategy.”

Response: We corrected the sentences accordingly.

Comment:

# 5 I am confused by the concluding sentence (p. 32, lines 335 to 337). What do the authors mean by “reducing the fragmentation” in the field of cognitive emotion regulation research? The meaning behind this statement should be explained.

Response: With “fragmentation" we refer to a situation in the field that we perceived as characterized by isolated research efforts and a lack of integration; we have tried to alleviate this with a replication-plus-extension approach in the design of this study. This final sentence, however, is probably not essential, and we have therefore deleted it to avoid further confusion.

Reviewer #2: 

Comment:

This fMRI study investigated the immediate and post-regulatory effects of emotion regulation via detachment in a sample of 48 healthy participants. Besides the emotion regulation phase, effects of emotion regulation were investigated for the post-regulation phase (directly after each trial), for re-exposure after 10 minutes and re-exposure after one week. The results especially show a time-dependent up- and down-regulation of amygdala activation in response to differing picture categories (negative/neutral/both) for the different phases. This is a highly relevant research question. In principle, the manuscript is well-written and the study procedure methodologically sound. However, the authors do not consequently distinguish between regulation of emotions (in response to negative pictures), detachment during neutral picture viewing or detachment during both picture types. Together with different analysis approaches for amygdala activation and varying results for the different picture categories, it is hard to understand the main message of the manuscript. I hope that the following comments might help to improve the manuscript.

Response: We thank the reviewer for their thorough, helpful and constructive comments, which are of great value for improving the manuscript. Below, we reply to the individual comments. Please note that, wherever we give references to pages or line numbers, these refer to the revised manuscript without track changes enabled.

Comment:

Introduction

1. The authors did not really differentiate between different reappraisal tactics even though there is evidence from previous studies regarding their differential effects (e.g. Dörfel et al., 2014; Hermann et al. 2020). I would appreciate the authors to elaborate on this issue.

Response: Thank you for this very important comment. Also following a comment from reviewer #1, we included the framework of {Powers, 2019, 30502352@@author-year} into the introduction in order to offer a definition, description and classification of detachment/distancing according to and extending the Process Model by {Gross, 1998, 9457784@@author-year}. As a consequence, we replaced the term “detachment” with “distancing” and introduced the distinction between “reinterpretation” and “distancing” as specific tactics of reappraisal in order to adapt our terminology to the framework. We also added explanations why distancing in particular was chosen for the present study. Additionally, we shortly describe the different forms of distancing and explain, which form we used in our investigation and why. Besides a clarification, this is just a change in wording, with no further implications for the results or their interpretation. We believe, however, that this is a step towards unified terminology and helps in comparing and aggregating results across studies. 

You will find the new paragraphs at page 3, line 45 ff.

Comment:

2. Page 5: Another example of long-term effects been established for episodic memory processes: -> incomplete sentence

Response: We corrected the sentence accordingly.

Comment: 3. This shouldbe accompanied by (page 6) -> should be

Response: We corrected the sentence accordingly.

Methods and materials

Comment:

4. The authors state that all data and material files are available from the Open Science Framework. I could however not find data from preprocessing or first-level models online. The authors should indicate which data are not provided and why, and/or add the respective data.

Response: We could only upload second-level results to the Open Science Framework due to space constraints. We will be able to provide first-level results at the repository of our local academic institution, https://opara.zih.tu-dresden.de/, and will add a link from the OSF repository to our local repository. Unfortunately, we could not do this in time due to repository maintenance and a mandatory reviewing process for our local repository, but we add the link at OSF as soon as possible. We changed the Data Availability Statement accordingly and added the following information: 

“Data and materials are provided at the Open Science Framework (https://osf.io/mg5ac/). Specifically, we provide data for first- and second-level fMRI analyses, the respective SPM code, ROI masks, behavioral data (arousal ratings, age, sex), the emotion regulation experiment, and R-code for reproducing the analyses and results. We are not able to provide raw and preprocessed fMRI data due to data privacy reasons.” 

Comment: 

5. How was the trial order during the emotion regulation task? This is not explicitly stated in the manuscript.

Response: Each set of pictures was arranged in two different sequences, which were assigned to participants in an alternating fashion. Within each sequence, the order of stimuli was pseudo-randomized with the constraint that all experimental conditions appeared equally within each experimental run, and that no more than three presentation of the same experimental condition occurred in succession. A brief description of the trial order has been added to the manuscript (see ‘Stimuli’ on pages 11 and 12).

Comment: 

6. Another issue concerns an aspect of the instruction during the re-exposure phase (“Specifically, they were instructed not to voluntarily change their emotional experience as they had done during the main experiment.”). Why did the authors decide to prevent the participants from spontaneously applying the prior regulation strategy during re-exposure (as a potential after-effect of emotion regulation)?

Response: We chose this instruction based on the Walter et al. {Walter, 2009, 21949675} study for which we aimed for maximal correspondence. The primary motivation for this instruction was as follows: in the re-exposure experiment, our goal was to investigate spontaneous emotional reactivity. We wanted to create comparable experimental conditions across participants and studies, and wanted to make clear to the participants that the task in the reactivity experiment was different from the task in the regulation experiment. For these reasons, we asked them not to voluntarily change their emotional experience. 

We acknowledge, though, that it is not entirely possible to prevent participants from applying spontaneous / involuntary emotion regulation with the chosen setup, and in fact, such regulation may be a part of an individual's response when encountering previous emotional stimuli. Our main goal was not to rule out such processes, but to ask participants to refrain from intentionally doing so. We now briefly discuss deviations from the instructions in the limitations section (pages 36 and 37).

Comment: 

7. The authors used three different types of strategies for analyzing amygdala activation. The authors should explicitly state that they did two different first-level analyses for the amygdala (transient/sustained response). This is not entirely clear for me.

Response: The use of two different first-level models has been clarified in the methods section (paragraph on fMRI analysis, page 13).

Comment: 

8. Did the authors use a high-pass filter?

Response: Yes, we used the default SPM high-pass filter (128s). This has been clarified in the methods section (paragraph on fMRI analysis, page 14).

Comment: 

9. How were the masks for the amygdala created in detail? Which probability threshold was used?

Response: We used the amygdala masks from the Harvard-Oxford Subcortical Structural Atlas. We chose this atlas due to its probabilistic nature. The probabilistic masks were thresholded at 50%. For analyses with these masks, we applied a threshold of p = .05 FWE after correction for small volume. The description in the manuscript has been adapted accordingly (paragraph on fMRI analysis, page 15).

Comment: 

10. The contrasts are not described in the methods section.

Response: We have added a description of the contrasts of the regulation task as well as the re-exposure task in the methods section (paragraph on fMRI analysis, page 13-14)

Results

Comment:

11. It is very time-consuming to understand the amygdala results which are spread over the main manuscript and the supplement. I suggest to integrate the main results (post-regulation, re-exposure) for all contrasts (at least for the amygdala) into the main manuscript. Moreover, it should explicitly be stated from which analyses the amygdala results stem (in Tables, Figures, main manuscript, supplement).

Response: We have moved the contents of the supplementary tables to the tables within the main manuscript. We changed the Figure and Table title/captions so that it is now indicated exactly to which point in time the results refer. Regarding the two analytic strategies for amygdala activation (transient and sustained responses), we have clarified in the manuscript (at the beginning of the results section) that we point out when the choice of the analytical model matters; if not indicated otherwise, results hold for both analytic strategies. Where necessary, we have made the results description more precise in this regard.

Discussion

Comment:

12. Throughout the abstract and discussion, it is not entirely clear for me which interpretation is related to which findings, especially regarding amygdala activation. If I understand it right, the immediate and short-/long-term regulation effects were partly found for negative pictures, neutral pictures or both picture categories (main effect). It is a very interesting result that amygdala activation is also decreased during detachment regarding neutral pictures. However, I do not understand why the authors interpret this effect as emotion regulation, as there are by definition no emotions elicited during neutral picture viewing. I would request the authors to elaborate on this issue. Moreover, it should be clearly stated throughout the manuscript (e.g. abstract, discussion) for which picture categories the effects have been found and what this means with regard to ‘emotion’ regulation or detachment irrespective of emotional content.

Response: We agree with the reviewer that the observation of decreased amygdala activation for neutral pictures during the detach condition as compared to the permit condition is interesting, and to some extent unexpected. It may be the case that it could not be detected in previous studies where no detach-neutral condition had been implemented. Our current interpretation is that the distinction between neutral and negative pictures may not be as clear cut as it seems, and might be more of a more/less nature. We speculate that the mere presentation of (bright) pictures in the dark environment of the scanner bore elicits an unspecific alerting/arousing response – and that this response can be the subject of emotional regulation. We have added a brief explanation to the discussion. 

We also restructured the discussion section a little bit, for instance by including sub-headings to make it clearer, which results are being discussed. By moving all supplementary tables into the main document, and labeling the tables and figures in a more exact way, we now hope, that reference to the respective results becomes easier. 

Comment: 

13. Moreover, the authors did not mention the result of increased right amygdala activation during detachment for negative compared with neutral pictures (emotion regulation phase) in the results section nor discussed this finding (see Table 1, last line). As enhanced amygdala activation for detachment > permit is also found for the short- (negative pictures) and long-term (main effect strategy; neutral pictures) re-exposure phase, I can’t understand why enhanced amygdala activation is only discussed for the re-exposure phases. How do the authors interpret this up-regulation of amygdala activation (sustained response) during detachment (emotion regulation phase) also in relation to the re-exposure findings?

Response: We agree that the interpretation of this particular interaction effect is difficult. The effect in question indicates a relative increase in right amygdala activation when using a boxcar model, i.e. that the difference between permit and detach (more precisely: permit minus detach) is greater for neutral than for negative stimuli. This is in contrast to the opposite interaction effect that was observed in both the left and right amygdala for the stick model.

We argue that for the interpretation of this interaction effect (i.e., a difference of differences) it is helpful to also consider the main effects. In contrast to the interaction effect in question, the main effects give a clear interpretation of greater amygdala activation for permit than detach (in all stimuli, in negative stimuli, and to a lesser extent also in neutral stimuli, observed in both the left and right amygdala and for both the stick and boxcar model). This interpretation is also supported by the analysis of extracted values, from which both absolute levels (to the extent that this is possible in fMRI) and relative differences can be identified. Taken together, we assume that the overall picture is that amygdala activation is decreased during detach as compared to permit. While other explanations may exist, we currently cannot think of a good explanation for the interaction effect in question, and tend to interpret this observation as a spurious result, because it is less consistently observed than the competing interaction effect. We now make the different patterns for transient and sustained responses more clear in the results section, and also mention the less consistent results for this effect in the discussion – which, as we believe, had already been toned down in this regard.

Regarding the second question, we are not sure if there is a relation of this effect to the increased activation during the re-exposure phase. While the effects appear to point into the same direction, they result from different experimental paradigms / contexts and different analytical models. For these reasons, we are reluctant to draw any conclusions about a potential relation.

Comment: 

14. The authors discuss the primarily transient response pattern within the amygdala (page 25). I do not understand on which results this interpretation is based, as there are both amygdala results for ROI and ROI/stick model.

Response: The reviewer is right that in the amygdala there are both results for the ROI and the ROI/stick model. Our interpretation of a primarily transient response pattern is based on the observation that for the ROI/stick model effects appeared to be more pronounced (with regard to statistical significance as well as spatial extent) and more consistent across hemispheres. During the regulation phase, for example, this is the case for the overall negative>neutral contrast and also for the various permit>detach contrasts. We concede, though, that this holds less for the regulation vs. post-regulation comparison, and not for the analysis of the post-regulation phase. In consequence, we have added an explanation for our interpretation to the discussion, and restrict the interpretation to the regulation phase.

Comment: 

15. The authors might discuss further differences to other studies/limitations of their study: the smaller sample size for re-exposure after one week, sex difference, lacking ratings during re-exposure,…

Response: We agree that there are a few more methodological limitations to our study, as the reviewer mentions. This concerns the smaller sample size and the lack of ratings during re-exposure as well as caveats regarding the adherence to instructions. We have now added those issues to the manuscript (see page 36, ‘Limitations’). Honestly speaking, we would not regard the non-investigation of sex differences as a limitation because this study was not designed with such a goal. It is nevertheless a worthwhile research question, but likely requires a larger sample size.

Plos One Editorial Office

Comment: 

Response: We have changed the style of all headings, some of the references that didn’t meet PLOS reference style requirements, Table as well as Figure captions and corrected the Figure file naming to meet the requirements.

Comment:

2. Please note that PLOS ONE does not copy edit accepted manuscripts (https://journals.plos.org/plosone/s/criteria-for-publication#loc-5). To that effect, please ensure that your submission is free of typos and grammatical errors.

Response: We checked very carefully during revision and implemented all points of the reviewers regarding typos and grammatical errors. 

Comment:

3. We noted in your submission details that a portion of your manuscript may have been presented or published elsewhere.

"No.

Results from the present sample on the research questions of this manuscript have not been reported previously nor are under consideration for publication elsewhere.

However, the presented study is part of a larger project of which several reports have already been published (Diers et al., 2014; Gärtner et al., 2019; Scheffel et al., 2019; Dörfel et al., 2020). Diers et al. (2014) reported results from a different sample on a different research question of the project, Gärtner et al. (2019), Scheffel et al. (2019), and Dörfel et al. (2020) combined data of this manuscript's sample with a third sample, but analyzed completely different research questions, each."

Please clarify whether this publication was peer-reviewed and formally published. If this work was previously peer-reviewed and published, in the cover letter please provide the reason that this work does not constitute dual publication and should be included in the current manuscript.

Response: Results from the present sample on the research questions of this manuscript have not been reported previously nor are under consideration for publication elsewhere. We believe that due to adherence to the pre-specified analysis plan and due to the non-overlap of research questions, which are too many and too diverse to address in a single manuscript, this work has an independent character and does not constitute dual publication. 

Please see the cover letter for a more detailed description. 

Comment: 

Response: This is no longer necessary because the supplementary figure is no longer needed and the supplementary tables have been incorporated to the main manuscript following some of the reviewer recommendations. Hence there is no Supporting Information any more. 

Plos One Editorial Office 2021-04-26

Comment:

1. Please ensure that the broader project from which the data originated is sufficiently described in the paper and referenced in your submission (and, if possible, included in the submission).

Response:

We added the following clarification to the manuscript (please see page 7, ‘Materials and methods’, page 6 in the original submission):

"Specifically, the project aimed at elucidating the effectiveness and potential costs of volitional emotion regulation. Different cognitive regulation strategies (acceptance, up- and down-regulation) were compared with respect to their behavioral and neural effectiveness, i.e., emotion regulation success, but operationalized along a prolonged time scale. This allowed to examine potential costs of volitional emotion regulation as indexed by paradoxical immediate and delayed regulatory after-effects. Further aims of the overall project were to investigate associations of emotion regulation success with personality traits and genetic polymorphisms. A detailed overview of the project and its subprojects is given in the Supplementary Materials (Fig S1). Therefore, some of the data reported in this article have been reused in three follow-up studies within the scope of this project: in accordance with the a priori specified analysis plan (http://gepris.dfg.de/gepris/projekt/223659428 and https://tu-dresden.de/bereichsuebergreifendes/sfb940/research/a-mechanismen/a5), associations with genetic polymorphisms were investigated [26], as well as the relation between emotion regulation and personality [27]. Additionally, associations of emotion regulation success and dispositional emotion regulation with resting-state cortico-limbic connectivity have been analyzed [18].” 

We also provide a detailed overview of the overall project together with the respective peer-reviewed publications that resulted from different arms of the project in Supplementary Fig S1.

---

## [Decision Letter · Decision Letter 1]

26 Jul 2021

Should we keep some distance from distancing? Regulatory and post-regulatory effects of emotion downregulation.

PONE-D-20-39022R1

Dear Dr. Dörfel,

We’re pleased to inform you that your manuscript has been judged scientifically suitable for publication and will be formally accepted for publication once it meets all outstanding technical requirements.

Kind regards,

Ilona Papousek

Academic Editor

PLOS ONE

Additional Editor Comments (optional):

Congratulations. Both expert reviewers noted that you have addressed all their requests and comments to their full satisfaction.

Reviewers' comments:

Reviewer's Responses to Questions

**Comments to the Author**

1. If the authors have adequately addressed your comments raised in a previous round of review and you feel that this manuscript is now acceptable for publication, you may indicate that here to bypass the “Comments to the Author” section, enter your conflict of interest statement in the “Confidential to Editor” section, and submit your "Accept" recommendation.

Reviewer #1: All comments have been addressed

Reviewer #2: All comments have been addressed

2. Is the manuscript technically sound, and do the data support the conclusions?

Reviewer #1: Yes

Reviewer #2: (No Response)

3. Has the statistical analysis been performed appropriately and rigorously? 

Reviewer #1: Yes

Reviewer #2: Yes

4. Have the authors made all data underlying the findings in their manuscript fully available?

Reviewer #1: Yes

Reviewer #2: Yes

5. Is the manuscript presented in an intelligible fashion and written in standard English?

Reviewer #1: Yes

Reviewer #2: Yes

6. Review Comments to the Author

Reviewer #1: (No Response)

Reviewer #2: (No Response)

7. PLOS authors have the option to publish the peer review history of their article (what does this mean?). If published, this will include your full peer review and any attached files.

Reviewer #1: No

Reviewer #2: No

---

## [Editor Report · Acceptance letter]

24 Aug 2021

PONE-D-20-39022R1 

Should we keep some distance from distancing? Regulatory and post-regulatory effects of emotion downregulation. 

Dear Dr. Dörfel:

I'm pleased to inform you that your manuscript has been deemed suitable for publication in PLOS ONE. Congratulations! Your manuscript is now with our production department. 

Kind regards, 

on behalf of

Dr. Ilona Papousek 

Academic Editor

PLOS ONE